# Boosting the Transferability of Adversarial Attacks with Reverse Adversarial Perturbation

## Abstract

Deep neural networks (DNNs) have shown to be vulnerable to adversarial examples, which can produce erroneous predictions by injecting imperceptible perturbations. In this work, we study the transferability of adversarial examples, which is of significant due to its threat to real-world applications where model architecture or parameters are usually unknown. Many existing works reveal that the adversarial examples are likely to overfit the surrogate model that they are generated from, limiting its transfer attack performance against different target models. Inspired by the connection between the flatness of loss landscape and the model generalization, we propose a novel attack method, dubbed *reverse adversarial perturbation* (RAP) to boost the transferability of adversarial examples. Specifically, instead of purely minimizing the adversarial loss at a single adversarial point, we advocate seeking adversarial examples locating at the low-value and flat region of the loss landscape, through injecting the worst-case perturbation (*i.e.*, the reverse adversarial perturbation) for each step of the optimization procedure. The adversarial attack with RAP is formulated as a min-max bi-level optimization problem. Comprehensive experimental comparisons demonstrate that RAP can significantly boost the adversarial transferability. Furthermore, RAP can be naturally combined with many existing black-box attack techniques, to further boost the transferability. When attacking a real-world image recognition system, *i.e.*, Google Cloud Vision API, we obtain 22% performance improvement of targeted attacks over the compared method.

## 1 Introduction

Deep neural networks (DNNs) have been successfully applied in many safety-critical tasks, such as autonomous driving, face recognition and verification, *etc*. However, it has been shown that DNN models are vulnerable to adversarial examples (Szegedy et al., 2013; Goodfellow et al., 2014), which are indistinguishable from natural examples but make a model produce erroneous predictions. For real-world applications, the DNN models are often hidden from users. Therefore, the attackers need to generate the adversarial examples under black-box setting where they do not know any information of the target model. For black-box setting, the adversarial transferability matters since it can allow the attackers to attack target models by using adversarial examples generated on the surrogate models. Therefore, learning how to generate adversarial examples with high transferability has gained more attentions in the literature (Liu et al., 2016; Tramèr et al., 2018; Dong et al., 2018; Xie et al., 2019b; Dong et al., 2019; Guo et al., 2020).

Under white-box setting where the complete information of the attacked model (*e.g.*, architecture and parameters) is available, the gradient-based attacks such as FGSM (Goodfellow et al., 2014) and I-FGSM (Kurakin et al., 2018) have demonstrated good attack performance. However, they often exhibit the poor transferiability (Xie et al., 2019b; Dong et al., 2018), *i.e.*, the adversarial examples generated from the surrogate models perform poorly against different target models. The previous works attribute that to the overfitting of adversarial examples to the surrogate models (Dong et al., 2018; Xie et al., 2019b; Lin et al., 2020). And various techniques have been proposed to improve the transferability, including input transformation (Dong et al., 2019; Xie et al., 2019b), gradient calibration (Guo et al., 2020), feature-level attacks (Huang et al., 2019), and generative models

(Naseer et al., 2021), etc. However, there still exists a large gap of attack performance between the transfer setting and the ideal white-box setting, especially for targeted attack, requiring more efforts for boosting the transferability.

Recalling that several existing works demonstrate that it is possible to enhance the standard/robust generalization of the learned model by seeking flat local minima in both standard and adversarial training of deep neural networks (Dziugaite & Roy, 2017; Li et al., 2018; Jiang et al., 2020; Izmailov et al., 2018; Foret et al., 2021; Wu et al., 2020b). We find that the transferability of adversarial examples between source and target models is somewhat analogous to the generalization of model parameters between training and testing sets: the formal aims to address the overfitting issue of the generated adversarial example to surrogate models, while the latter tends to alleviate the overfitting of the model parameters to the training set. The perspective motivates us to think about the following question: *can we alleviate the overfitting to surrogate models by investigating the flatness of loss landscape around the generated adversarial examples?*

In this work, we propose a novel attack method called *reverse adversarial perturbation* (RAP) to boost the transferability of adversarial examples. The key idea is to seek an adversarial example that is not only of low adversarial loss but also locates at a local flat region, *i.e.*, the points within the local neighborhood region around this adversarial example should also of similar low loss values. To achieve this goal, we design a min-max bi-level optimization problem. The inner maximization aims to find the worst-case perturbation (*i.e.*, that with the largest adversarial loss, and this is why we call it reverse adversarial perturbation) within the local region around the current adversarial example, which can be solved by the projected gradient ascent algorithm. Then, the outer minimization will update the adversarial example to find a new point added with the provided reverse perturbation that leads to lower adversarial loss. Besides, we design a late-start variant of RAP (RAP-LS) to further boost the attack effectiveness and efficiency, which doesn't insert the reverse perturbation into the optimization procedure in the early stage. Extensive experiments on attacking several DNN models have empirically verified that the adversarial examples generated by the proposed RAP method often locate at flat regions, and show much higher adversarial transferability, compared to existing methods. Moreover, from the technical perspective, since the proposed RAP method only introduces one specially designed perturbation onto the adversarial example, one notable advantage of the proposed method is that it can be naturally combined with many existing black-box attack techniques to further boost the transferability. For example, when combined with different input transformations (*e.g.*, the random resizing and padding in Diverse Input (Xie et al., 2019b)), our RAP method consistently outperforms the counterparts by a clear margin.

Our main contributions are four-fold: 1) we advocate considering the flatness of loss landscape around the adversarial example to boost the adversarial tranferability; 2) we propose the *reverse adversarial perturbation* (RAP) attack to explicitly encourage the local flatness around the adversarial example by injecting the worst-case perturbation; 3) we present a vigorous experimental study and show that RAP can significant boost the adversarial transferability on both untargeted and targeted attacks; 4) we demonstrate that RAP can be easily combined with existing transfer attack techniques and outperforms the state-of-the-art performance by a large margin.

## 2 RELATED WORK

The black-box attacks can be categorized into two categories: 1) *query-based attacks* that conduct the attack based on the feedback of iterative queries to target models, and 2) *transfer attacks* that use the adversarial examples generated on some surrogate models to attack the target models. In this work, we focus on the transfer attacks. For surrogate models, existing attack algorithms such as FGSM (Goodfellow et al., 2014) and I-FGSM (Kurakin et al., 2018) could achieve good attack performance. However, they often overfit the surrogate models and thus exhibit poor transferability. Recently, many works are proposed to generate more transferable adversarial examples.

Instead of the I-FGSM, the work of Dong et al. (2018) integrate momentum into the updating strategy and Lin et al. (2020) use the Nesterov accelerated gradient to boost the transferability. Data augmentation, which has been shown to be effective in improving model generalization, has also been widely studied in transfer attack, such as randomly resizing and padding (Xie et al., 2019b), randomly scaling (Lin et al., 2020), and adversarial mixup (Wang et al., 2021b). The work of Dong et al. (2019) use a set of translated images to compute gradient and get the better performance against defense models. There are also some model-specific designs to boost the transferability. For

example, Wu et al. (2020a) find the gradient of skip connections is more crucial to generate more transferable attacks. The work of Guo et al. (2020) also propose LinBP to utilize more gradient of skip connections during the back-propagation. However, these methods tend to be specific to a particular model architecture, such as skip connection, and it is nontrivial to extend the findings to other architectures or modules. Meanwhile, Huang et al. (2019); Inkawhich et al. (2020a;b) propose to exploit feature space constraints to generate more transferable attacks. Yet they need to identity the best performing intermediate layers or train one-vs-all binary classifies for all attacked classes. Recently, Zhao et al. (2020) find iterative attacks with much more iterations and logit loss can achieve relatively high targeted transferability and exceed the feature-based attacks.

Apart from the input-specific adversarial attacks, there have been some methods utlizing the generative models to generate the adversarial perturbations (Poursaeed et al., 2018; Naseer et al., 2019; 2021). For example, the work of Naseer et al. (2021) propose to train a generative model to match the distributions of source and target class, so as to increase the targeted transferability. However, the learning of the perturbation generator is nontrivial, especially on large-scale datasets. In summary, the current performance of transfer attacks is still unsatisfactory, especially for targeted attacks.

Wang et al. (2021a) also explore the min-max framework for producing the attacks, but they are totally different from us in terms of motivation and formulation. They are to generate attacks based on multiple models and samples. And, the variable of inner maximization in Wang et al. (2021a) is the probability vector, representing the weight assigned to different models or samples.

## 3 METHODOLOGY

### 3.1 PRELIMINARIES OF TRANSFER ADVERSARIAL ATTACK

Given an benign sample $(\boldsymbol{x}, y) \in (\mathcal{X}, \mathcal{Y})$, the procedure of transfer adversarial attack is firstly constructing the adversarial example $\boldsymbol{x}^{adv}$ within the neighborhood region $\mathcal{B}_\epsilon(\boldsymbol{x}) = \{\boldsymbol{x}' : \|\boldsymbol{x}' - \boldsymbol{x}\|_p \leq \epsilon\}$ by attacking the white-box surrogate model $\mathcal{M}^s(\boldsymbol{x}; \boldsymbol{\theta}) : \mathcal{X} \to \mathcal{Y}$, then transferring $\boldsymbol{x}^{adv}$ to directly attack the black-box target model $\mathcal{M}^t(\boldsymbol{x}; \boldsymbol{\phi}) : \mathcal{X} \to \mathcal{Y}$. The attack goal is to mislead the target model, *i.e.*, $\mathcal{M}^t(\boldsymbol{x}^{adv}; \boldsymbol{\phi}) \neq y$ (untargeted attack), or $\mathcal{M}^t(\boldsymbol{x}^{adv}; \boldsymbol{\phi}) = y_t$ (targeted attack) with $y_t \in \mathcal{Y}$ indicting the target label. Taking the target attack as example, the general formulation of many existing transfer attack methods can be written as follows:

$$\min_{\boldsymbol{x}^{adv} \in \mathcal{B}_\epsilon(\boldsymbol{x})} \mathcal{L}(\mathcal{M}^s(\mathcal{G}(\boldsymbol{x}^{adv}); \boldsymbol{\theta}), y_t). \tag{1}$$

The loss function $\mathcal{L}$ is often set as the cross entropy (CE) loss (Xie et al., 2019b) or the logit loss (Zhao et al., 2020), which will be specified in later experiments. Besides, the formulation of untargeted attack can be easily obtained by replacing the loss function $\mathcal{L}$ and $y_t$ by $-\mathcal{L}$ and $y$, respectively.

Since $\mathcal{M}^s$ is white-box, if $\mathcal{G}(\cdot)$ is set as the identity function, then any off-the-shelf white-box adversarial attack method can be adopted to solve Problem (1), such as I-FSGM (Kurakin et al., 2018), MI-FGSM (Dong et al., 2018), *etc*. Meanwhile, some existing works have designed different $\mathcal{G}(\cdot)$ functions and developed the corresponding optimization algorithms, in order to boost the adversarial transferability between the surrogate and target models. For example, $\mathcal{G}(\cdot)$ is specified as random resizing and padding (DI) (Xie et al., 2019b), translation transformation (TI) (Dong et al., 2019), scale transformation (SI) (Lin et al., 2020), and adversarial mixup (Admix) (Wang et al., 2021b).

### 3.2 TRANSFER ADVERSARIAL ATTACK WITH REVERSE ADVERSARIAL PERTURBATION

Adversarial transferability between surrogate to target models is somewhat analogous to the model generalization between training and testing sets. The former studies the overfitting to the surrogate model when constructing an adversarial example, and the latter focuses on the overfitting to the training set when training a model. Inspired by this point, some previous works have attempted to borrow the effective techniques on improving model generalization to enhance the adversarial transferability. For example, inspired by data augmentation, a series of transfer attack methods focused on augmenting the adversarial perturbation during the procedure of generating adversarial perturbation, and showed good transferability, such as DI (Xie et al., 2019b), TI (Dong et al., 2019), SI (Lin et al., 2020), and Admix (Wang et al., 2021b).

Recalling that prior works have related the model generalization to the flatness (or sharpness) of the loss landscape in the weight space, motivating a series of methods on boosting the model generalization by implicitly or explicitly seeking a flatter minima (Dziugaite & Roy, 2017; Li et al., 2018;

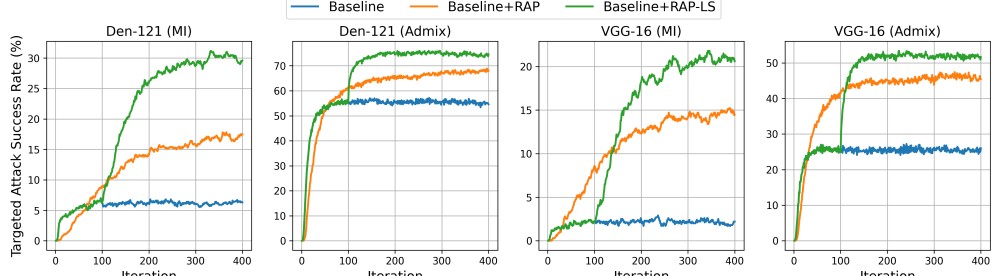

Figure 1: Targeted attack success rate ($\%$) on Dense-121 and VGG-16. We take the Res-50 as the surrogate model and take MI and Admix as baseline methods.

Jiang et al., 2020; Izmailov et al., 2018; Foret et al., 2021). Inspired by this point, we assume that the adversarial example locating at flatter regions of the loss landscape of the surrogate model may have better adversarial transferability to other models. Specifically, we encourage that not only $\boldsymbol{x}^{adv}$ itself has low loss value, but also the points in the vicinity of $\boldsymbol{x}^{adv}$ have similarly low loss values. To achieve this goal, we propose to minimize the maximal loss value within a local neighborhood region around the adversarial example $\boldsymbol{x}^{adv}$. The maximal loss is implemented by perturbing $\boldsymbol{x}^{adv}$ to maximize the adversarial loss, named *Reverse Adversarial Perturbation* (RAP). By inserting the RAP into the formulation (1), we aim to solve the following problem,

$$\min_{\boldsymbol{x}^{adv}\in\mathcal{B}_\epsilon(\boldsymbol{x})} \mathcal{L}(\mathcal{M}^s(\mathcal{G}(\boldsymbol{x}^{adv}+\boldsymbol{n}^{adv});\boldsymbol{\theta}),y_t), \tag{2}$$

where

$$\boldsymbol{n}^{adv} = \argmax_{\|\boldsymbol{n}^{adv}\|_\infty \leq \epsilon_n} \mathcal{L}(\mathcal{M}^s(\boldsymbol{x}^{adv}+\boldsymbol{n}^{adv};\boldsymbol{\theta}),y_t), \tag{3}$$

with $\boldsymbol{n}^{adv}$ indicating the RAP, and $\epsilon_n$ defining its search region. The above formulations Equation (2) and Equation (3) correspond to the targeted attack, and the corresponding untargeted formulations can be easily obtained by replacing the loss function $\mathcal{L}$ and $y_t$ by $-\mathcal{L}$ and $y$, respectively.

It is a min-max bi-level optimization problem (Liu et al., 2021), and can be solved by iteratively optimizing the inner maximization and the outer minimization problem. Specifically, in each iteration, given $\boldsymbol{x}^{adv}$, the inner maximization *w.r.t.* $\boldsymbol{n}^{adv}$ is solved by the projected gradient ascent algorithm:

$$\boldsymbol{n}^{adv} \leftarrow \boldsymbol{n}^{adv} + \alpha_n \cdot \mathrm{sign}(\nabla_{\boldsymbol{n}^{adv}}\mathcal{L}(\mathcal{M}^s(\boldsymbol{x}^{adv}+\boldsymbol{n}^{adv};\boldsymbol{\theta}),y_t)). \tag{4}$$

The above update is conducted by $T$ steps, and $\alpha_n = \frac{\epsilon_n}{T}$. Then, given $\boldsymbol{n}^{adv}$, the outer minimization *w.r.t.* $\boldsymbol{x}^{adv}$ can be solved by any off-the-shelf algorithm that is developed for solving Equation (1). For example, it can be undated by one step projected gradient descent, as follows:

$$\boldsymbol{x}^{adv} \leftarrow \mathrm{Clip}_{\mathcal{B}_\epsilon(\boldsymbol{x})}\big[\boldsymbol{x}^{adv} - \alpha \cdot \mathrm{sign}(\nabla_{\boldsymbol{x}^{adv}}\mathcal{L}(\mathcal{M}^s(\mathcal{G}(\boldsymbol{x}^{adv}+\boldsymbol{n}^{adv});\boldsymbol{\theta}),y_t))\big], \tag{5}$$

with $\mathrm{Clip}_{\mathcal{B}_\epsilon(\boldsymbol{x})}(\boldsymbol{a})$ clipping $\boldsymbol{a}$ into the neighborhood region $\mathcal{B}_\epsilon(\boldsymbol{x})$. The overall optimization procedure is summarized in Algorithm 1. Since the optimization *w.r.t.* $\boldsymbol{x}^{adv}$ can be implemented by any off-the-shelf algorithm for solving Problem Equation (1), one notable advantage of the proposed RAP is that it can be naturally combined with any one of them, such as the input transformation methods (Xie et al., 2019b; Dong et al., 2019; Lin et al., 2020; Wang et al., 2021b).

**A Late-Start (LS) Variant of RAP.** In our preliminary experiments, we find that RAP requires more iterations to converge and the performance is slightly lower during the initial iterations, compared to its baseline transfer attack methods. As shown in Figure 1, we combine MI (Dong et al., 2018) and Admix (Wang et al., 2021b) with RAP, and adopt ResNet-50 as the surrogate model. We take the evaluation on 1000 images from ImageNet (see Sec.4.1). It is observed that the method with RAP (see the orange curves) quickly surpasses its baseline method (see the blue curves) and finally achieves much higher success rate with more iterations, which verify the effect of RAP on enhancing the adversarial transferability. However, it is also observed that the performance of RAP is slightly lower than its baseline method in the early stage. The possible reason is that the early-stage adversarial example is of very weak attack performance to the surrogate model. In this case, it may be waste to pursue better transferable adversarial example by solving the min-max problem. A better strategy may be only solving the minimization problem Equation (1) in the early stage to

---

**Algorithm 1** Transfer Adversarial Attack Algorithm with Reverse Adversarial Perturbation (RAP)

---

**Input:** Surrogate model $\mathcal{M}^s$, benign data $(\boldsymbol{x}, y)$, target label $y_t$, loss function $\mathcal{L}$, transformation $\mathcal{G}$, the global iteration number $K$, the late-start iteration number $K_{LS}$ of RAP, as well as hyper-parameters in optimization (specified in later experiments)
**Output:** the adversarial example $\boldsymbol{x}^{adv}$

1: **Initialize** $\boldsymbol{x}^{adv} \leftarrow \boldsymbol{x}$
2: **for** $k = 1, \ldots, K$ **do**
3:   **if** $k \geq K_{LS}$ **then**
4:     **Initialize** $\boldsymbol{n}^{adv} \leftarrow \boldsymbol{0}$
5:     **for** $t = 1, \ldots, T$ **do**
6:       Update $\boldsymbol{n}^{adv}$ using Equation (4)
7:   Update $\boldsymbol{x}^{adv}$ using Equation (5)

---

quickly achieve the region of relatively high adversarial attack performance, then starting RAP to further enhance the attack performance and transferability simultaneously. This strategy is denoted as RAP with late-start (RAP-LS), whose effect is preliminarily supported by the results shown in Figure 1 (see the green curve) and will be evaluated extensively in later experiments.

## 4 EXPERIMENTS

### 4.1 EXPERIMENTAL SETTINGS

**Dataset and Evaluated Models.** We conduct the evaluation on the ImageNet-compatible dataset [1] comprised of 1,000 images. For the surrogate models, we consider the four widely used network architectures: Inception-v3 (Inc-v3) (Szegedy et al., 2016), ResNet-50 (Res-50) (He et al., 2016), DenseNet-121 (Dense-121) (Huang et al., 2017), and VGG-16bn (VGG-16) (Simonyan & Zisserman, 2014). For target models, apart from the above models, we also utilize more diverse architectures: Inception-ResNet-v2 (Inc-Res-v2) (Szegedy et al., 2017), NASNet-Large (NASNet-L) (Zoph et al., 2018), and ViT-Base/16 (ViT-B/16) (Dosovitskiy et al., 2021). For defense models, we adopt the two widely used adversarial trained models: adv-Inc-v3 (Inc-v3$_{adv}$) and ens-adv-Inc-Res-v2 (IncRes-v2$_{ens}$) (Tramèr et al., 2018).

**Compared Methods.** We adopt I-FGSM (Kurakin et al., 2018) (denoted as I), MI (Dong et al., 2018), TI (Dong et al., 2019), DI (Xie et al., 2019b), SI (Lin et al., 2020), Admix (Wang et al., 2021b), ILA (Huang et al., 2019), LinBP (Guo et al., 2020), and the generative targeted attack method TTP (Naseer et al., 2021). We also consider the combination of baseline methods, including MI-TI-DI (MTDI), MI-TI-DI-SI (MTDSI), and MI-TI-DI-Admix (MTDAI).

**Implementation Details.** For untargeted attack, we adopt the Cross Entropy (CE) loss. For targeted attack, apart from CE, we also experiment with the logit loss, where Zhao et al. (2020) shows it behaves better for targeted attack. The adversarial perturbation $\epsilon$ is restricted by $\ell_\infty = 16/255$. The step size $\alpha$ is set as $2/255$ and number of iteration $K$ is set as $400$ for all attacks. In the following, we mainly show the results at $K = 400$ and the results at different value of $K$ are shown in *supplementary materials*. For RAP, we set $K_{LS}$ as $100$ and $\alpha_n$ as $2/255$. We set $\epsilon_n$ as $12/255$ for I and TI in untargeted attack and $16/255$ for other attacks in all other settings.

### 4.2 A CLOSER LOOK AT THE FLATNESS OF RAP

We first conduct experiments to study the effect of RAP on the flatness of the loss landscape around the adversarial examples. We use ResNet-50 as surrogate model and conduct the targeted attacks. We take I, MI, DI, and MTDI as baselines and combined them with RAP. We visualize the flatness of the loss landscape around $\boldsymbol{x}^{adv}$ on surrogate model by plotting the loss variations when we move $\boldsymbol{x}^{adv}$ along a random direction with different magnitudes $a$. The details of the calculation are provided in *supplementary materials*. Figure 2 plots the visualizations. We can see that comparing to the baselines, RAP significantly improves the flatness of loss landscape around $\boldsymbol{x}^{adv}$. In the following, we investigate whether the flatness induced by RAP can boost the transferability of adversarial examples or not, on both untargeted and targeted transfer attacks.

---

[1]Publicly available from `https://github.com/cleverhans-lab/cleverhans/tree/master/cleverhans_v3.1.0/examples/nips17_adversarial_competition/dataset`

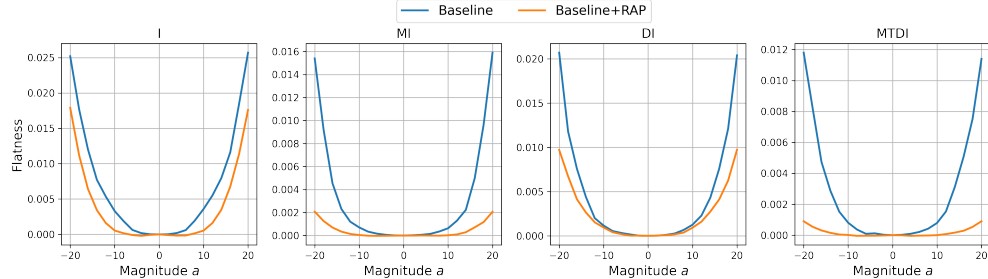

Figure 2: The flatness visualization of targeted adversarial examples.

Table 1: The **untargeted attack success rate** (%) **of baseline attacks with RAP**. The results with $CE$ loss are reported. The best results are bold and the second best results are underlined.

| Attack | **ResNet-50** $\Longrightarrow$ | | | **DenseNet-121** $\Longrightarrow$ | | |
|---|---|---|---|---|---|---|
| | Dense-121 | VGG-16 | Inc-v3 | Res-50 | VGG-16 | Inc-v3 |
| I / +RAP / +RAP-LS | 79.2 / 91.5 / **91.9** | 78.0 / 91.1 / **92.9** | 34.6 / 57.0 / **57.2** | 87.4 / 94.2 / **94.3** | 85.1 / 91.7 / **92.8** | 46.5 / 60.2 / **61.1** |
| MI / +RAP / +RAP-LS | 85.8 / 95.0 / **96.1** | 82.4 / 93.9 / **94.5** | 50.3 / 75.9 / **77.4** | 90.3 / 97.6 / **97.6** | 87.5 / 96.0 / **97.6** | 59.3 / 80.4 / **82.8** |
| TI / +RAP / +RAP-LS | 82.0 / 94.1 / **95.1** | 81.0 / 93.1 / **93.3** | 45.5 / 66.1 / **67.0** | 89.6 / 94.2 / **94.8** | 87.0 / 92.1 / **93.3** | 54.2 / 66.7 / **70.0** |
| DI / +RAP / +RAP-LS | 99.0 / 99.6 / **99.7** | 99.0 / 99.6 / **99.7** | 57.7 / 82.9 / **85.0** | 98.2 / 99.6 / **99.7** | 98.1 / **99.4** / **99.4** | 67.6 / 86.6 / **86.9** |
| SI / +RAP / +RAP-LS | 94.9 / 98.9 / **99.7** | 88.6 / 95.7 / **97.2** | 65.9 / 79.7 / **84.4** | 95.1 / 96.9 / **98.8** | 91.9 / 95.0 / **97.5** | 71.6 / 83.2 / **87.4** |
| Admix / +RAP / +RAP-LS | 97.9 / 99.6 / **99.9** | 95.8 / 97.7 / **99.0** | 77.7 / 87.4 / **92.6** | 97.0 / 99.0 / **99.2** | 95.6 / 97.7 / **98.6** | 82.0 / 89.8 / **93.8** |

| Attack | **VGG-16** $\Longrightarrow$ | | | **Inc-v3** $\Longrightarrow$ | | |
|---|---|---|---|---|---|---|
| | Res-50 | Dense-121 | Inc-v3 | Res-50 | Dense-121 | VGG-16 |
| I / +RAP / +RAP-LS | 53.7 / 53.0 / **54.2** | 49.1 / 50.6 / **51.4** | 22.0 / 24.7 / **24.9** | 51.5 / **62.1** / 62.0 | 48.7 / **60.8** / 60.0 | 55.1 / 65.9 / **68.0** |
| MI / +RAP / +RAP-LS | 62.5 / 76.2 / **76.4** | 60.5 / 73.0 / **73.9** | 30.0 / **42.7** / 42.2 | 62.0 / **85.8** / 84.8 | 56.7 / **84.6** / 84.6 | 63.1 / **84.9** / 84.6 |
| TI / +RAP / +RAP-LS | 62.8 / 64.8 / **65.8** | 55.9 / **63.7** / 62.1 | 29.1 / 36.2 / **37.1** | 49.3 / **63.4** / 61.6 | 49.4 / 63.4 / **63.8** | 58.1 / 68.6 / **69.5** |
| DI / +RAP / +RAP-LS | 72.2 / 86.0 / **88.8** | 68.8 / 85.0 / **87.4** | 29.9 / 46.6 / **51.6** | 68.4 / 81.7 / **81.8** | 71.9 / **85.0** / 84.0 | 76.1 / 85.2 / **86.4** |
| SI / +RAP / +RAP-LS | 80.0 / 92.7 / **94.7** | 82.1 / 94.8 / **95.7** | 45.8 / 74.0 / **74.7** | 66.2 / 69.8 / **72.8** | 65.9 / 74.9 / **77.2** | 66.0 / 69.2 / **73.0** |
| Admix / +RAP / +RAP-LS | 87.3 / 94.6 / **96.8** | 88.2 / 96.4 / **97.2** | 55.5 / 77.6 / **80.8** | 75.9 / 80.2 / **84.9** | 78.5 / 83.7 / **87.4** | 74.5 / 77.2 / **83.5** |

## 4.3 THE EVALUATION OF UNTARGETED ATTACKS

**Baseline Methods.** We first evaluate the performance of RAP and RAP-LS with different baseline attacks, including I, MI, DI, TI, SI, and Admix. The results are shown in Table 1. For instance, the 'MI/ +RAP/ +RAP-LS' denotes the methods of baseline MI, MI+RAP, and MI+RAP-LS, respectively. RAP achieves the significant improvements for all methods on each target model. For average attack success rate of all target models, RAP outperforms the I and MI by 9.6% and 16.3%, respectively. For TI, DI, SI, and Admix, RAP gets the improvements by 10.2%, 10.9%, 9.3%, and 6.3%. With late-start, RAP-LS further enhance the transfer attack performance for almost all methods.

**Combinational Methods.** Prior works demonstrate the combination of baseline methods could largely boost the adversarial transferability (Zhao et al., 2020; Wang et al., 2021b). We also investigate of behavior of RAP when incorporated with the combinational attacks. The results are shown in Table 2. As shown in the table, there exist the clear improvements of the combinational attacks over all baseline attacks shown in Table 1. In addition, our RAP-LS further boosts the average attack success rate of the three combinational attacks by 6.9%, 2.6%, and 1.7% respectively. Combined with the three combinational attacks, RAP-LS achieves 95.4%, 97.6%, and 98.3% average attack success rate, respectively. These results demonstrate RAP can significantly enhance the transferability.

## 4.4 THE EVALUATION OF TARGETED ATTACKS

We then evaluate the targeted attack performance of the different methods with RAP. The results with logit loss are presented and the results with CE loss are shown in *supplementary materials*.

**Baseline Methods.** The results of RAP with baseline attacks are shown in Table 3. From the results, RAP is also very effective in enhancing the transferability in targeted attacks. Taking ResNet-50 and DenseNet-121 as surrogate models for example, the average performance improvements induced by RAP are 5.0% (I), 8.1% (MI), 4.6% (TI), 10.4% (DI), 18.5% (SI), and 15.1% (Admix), respectively. Comparing to the ResNet-50 and DenseNet-121, the baseline attacks generally achieve lower transferability when using the VGG-16 or Inception-v3 as the surrogate models, which has also been verified in existing works (Xie et al., 2019b; Zhao et al., 2020). However, for Inception-v3 and VGG-16 as the surrogate models, RAP also consistently boosts the transferability under all cases. With late-start, RAP-LS could further improve the transferability of RAP for most attacks. The average attack success rate under all attack cases of RAP-LS is 2.6% higher than that of RAP.

Table 2: The **untrageted attack success rate** (%) **of combinational methods with RAP**. The results with $CE$ loss are reported. The best results are bold and the second best results are underlined.

| Attack | ResNet-50 $\Longrightarrow$ | | | DenseNet-121 $\Longrightarrow$ | | |
|---|---|---|---|---|---|---|
| | Dense-121 | VGG-16 | Inc-v3 | Res-50 | VGG-16 | Inc-v3 |
| MTDI / +RAP / +RAP-LS | 99.8 / **100** / **100** | 99.8 / **100** / 99.9 | 85.7 / **96.0** / 96.9 | 99.4 / 99.8 / **100** | 99.2 / 99.5 / **100** | 89.1 / **97.1** / **97.1** |
| MTDSI / +RAP / +RAP-LS | **100** / **100** / **100** | 99.7 / **99.9** / 99.8 | 97.0 / **99.1** / 99.1 | 99.8 / **99.9** / 99.9 | 99.2 / 99.3 / **99.7** | 95.1 / 98.3 / **98.4** |
| MTDAI / +RAP / +RAP-LS | **100** / **100** / **100** | 99.8 / **99.9** / 99.9 | 98.3 / 99.2 / **99.8** | 99.8 / **99.8** / 99.9 | 99.4 / 99.6 / **99.8** | 97.9 / 98.8 / **98.9** |

| Attack | VGG-16 $\Longrightarrow$ | | | Inc-v3 $\Longrightarrow$ | | |
|---|---|---|---|---|---|---|
| | Res-50 | Dense-121 | Inc-v3 | Res-50 | Dense-121 | VGG-16 |
| MTDI / +RAP / +RAP-LS | 90.0 / **97.2** / 97.7 | 88.8 / **97.0** / 97.3 | 56.8 / **82.6** / 81.4 | 82.9 / **91.8** / 90.6 | 85.7 / **94.2** / 93.3 | 85.1 / **92.7** / 91.0 |
| MTDSI / +RAP / +RAP-LS | 97.6 / **98.8** / 99.4 | 98.1 / 99.2 / **99.4** | 85.0 / 94.1 / **95.2** | 89.0 / 91.2 / **92.3** | 92.0 / 95.2 / **95.6** | 87.6 / 90.3 / **92.2** |
| MTDAI / +RAP / +RAP-LS | 97.8 / **99.2** / 99.6 | 98.9 / 99.5 / **99.6** | 89.3 / 95.0 / **95.5** | 91.5 / 94.1 / **94.7** | 95.4 / 96.2 / **97.6** | 91.4 / 93.2 / **94.1** |

Table 3: The **targeted attack success rate** (%) **of baseline methods with RAP**. The results with logit loss are reported. The best results are bold and the second best results are underlined.

| Attack | ResNet-50 $\Longrightarrow$ | | | DenseNet-121 $\Longrightarrow$ | | |
|---|---|---|---|---|---|---|
| | Dense-121 | VGG-16 | Inc-v3 | Res-50 | VGG-16 | Inc-v3 |
| I / +RAP / +RAP-LS | 4.5 / 9.5 / **14.3** | 2.4 / 9.8 / **11.8** | 0.1 / 0.1 / **0.7** | 5.0 / 12.8 / **17.9** | 2.9 / 10.1 / **15.9** | 0.0 / 0.8 / **1.2** |
| MI / +RAP / +RAP-LS | 6.3 / 17.5 / **29.6** | 2.2 / 14.5 / **24.6** | 0.1 / 1.1 / **2.4** | 4.6 / 16.2 / **26.5** | 3.1 / 13.4 / **23.2** | 0.3 / 2.0 / **3.4** |
| TI / +RAP / +RAP-LS | 7.2 / 11.0 / **17.3** | 4.0 / 12.9 / **15.3** | 0.1 / 0.8 / **1.2** | 8.4 / 13.5 / **20.8** | 5.2 / 12.4 / **16.4** | 0.2 / 2.1 / **3.0** |
| DI / +RAP / +RAP-LS | 62.6 / 64.9 / **73.9** | 57.2 / 63.4 / **69.3** | 1.5 / 7.9 / **10.1** | 30.2 / 52.6 / **60.4** | 32.1 / 49.5 / **58.9** | 1.4 / 8.8 / **10.0** |
| SI / +RAP / +RAP-LS | 30.0 / 53.2 / **61.1** | 9.5 / 32.8 / **36.0** | 1.8 / 9.3 / **10.5** | 14.2 / 41.5 / **43.4** | 8.4 / 31.0 / **35.2** | 1.6 / 8.5 / **10.4** |
| Admix / +RAP / +RAP-LS | 54.6 / 68.0 / **74.6** | 26.0 / 45.4 / **51.6** | 5.8 / 17.1 / **19.6** | 29.3 / 53.0 / **58.2** | 21.5 / 42.7 / **48.2** | 5.0 / 17.1 / **17.6** |

| Attack | VGG-16 $\Longrightarrow$ | | | Inc-v3 $\Longrightarrow$ | | |
|---|---|---|---|---|---|---|
| | Res-50 | Dense-121 | Inc-v3 | Res-50 | Dense-121 | VGG-16 |
| I / +RAP / +RAP-LS | 0.1 / 0.7 / **1.4** | 0.2 / 1.4 / **1.7** | 0.0 / 0.1 / **0.2** | 0.2 / **0.9** / 0.5 | 0.2 / **0.6** / 0.3 | 0.1 / **0.5** / 0.5 |
| MI / +RAP / +RAP-LS | 0.5 / 1.3 / **1.9** | 0.5 / 2.3 / **3.0** | 0.0 / 0.0 / **0.3** | 0.2 / **1.6** / 1.5 | 0.1 / 1.6 / **1.5** | 0.2 / 1.3 / **1.0** |
| TI / +RAP / +RAP-LS | 0.7 / 1.2 / **3.2** | 0.8 / 1.7 / **2.9** | 0.0 / 0.1 / **0.4** | 0.2 / 0.5 / **0.7** | 0.1 / 0.7 / **0.6** | 0.2 / 0.8 / **0.6** |
| DI / +RAP / +RAP-LS | 2.8 / 7.3 / **9.7** | 3.8 / 8.4 / **12.7** | 0.0 / 0.4 / **1.1** | 1.6 / 4.6 / **6.4** | 2.8 / 5.8 / **7.5** | 2.6 / 6.3 / **8.1** |
| SI / +RAP / +RAP-LS | 3.3 / 9.8 / **9.8** | 7.2 / 16.8 / **17.8** | 0.2 / 1.7 / **1.8** | 0.6 / 2.9 / **2.5** | 0.9 / 2.7 / **3.2** | 0.5 / 1.5 / **2.3** |
| Admix / +RAP / +RAP-LS | 5.6 / 11.1 / **11.9** | 13.0 / 20.2 / **23.6** | 0.7 / 2.4 / **2.8** | 1.5 / 4.9 / **5.2** | 2.0 / 6.9 / **7.5** | 1.3 / 3.3 / **4.4** |

**Combinational Methods.** As did in the untargeted attacks, we also evaluate the performance of combinational methods. The results are shown in Table 4. Similar to the findings in untargeted attacks, the combinational methods obtain significantly improvements over baseline methods. The RAP-LS outperforms all combinational methods by a significantly margin. For example, taking the average attack success rate of all target models as evaluation metric, RAP-LS obtains 14.2%, 11.8%, 9.3% improvements over the MTDI, MTDSI and MTDAI, respectively.

### 4.5 THE COMPARISON WITH OTHER TYPES OF ATTACKS

Apart from the baseline and the combinational methods, we also experiment with more diverse attack methods, including the model-specific attack LinBP (Guo et al., 2020), the feature-based attack ILA (Huang et al., 2019), and the generative targeted attack TTP (Naseer et al., 2021).

The LinBP depends on the skip connection and the authors only provide the source code about ResNet-50. We use their released code and thus conduct experiments with ResNet-50 as surrogate model. The results of LibBP and ILA are shown in Table 5, where we also implement the variants of LinBP following Guo et al. (2020), inlcuding LinBP-ILA, LinBP-ILA-SGM, LinBP-MI-DI, and LinBP-MI-DI-SGM. We observe that our MI-DI-RAP significantly outperforms the LinBP and ILA, especially for the targeted attacks. Compared with the second-best method (*i.e.*, LinBP-MI-DI-SGM), we obtain a large improvement by 33.5% on average ASR of targeted attacks.

Table 5: The comparison with ILA and LinBP. We use ResNet-50 as surrogate model. The best results are bold.

| Attack | Untarged | | | Targeted | | |
|---|---|---|---|---|---|---|
| | Dense-121 | VGG-16 | Inc-v3 | Dense-121 | VGG-16 | Inc-v3 |
| ILA | 95.0 | 94.2 | 77.7 | 2.8 | 1.5 | 0.5 |
| LinBP-ILA | 99.5 | 99.2 | 89.8 | 9.4 | 4.9 | 2.0 |
| LinBP-ILA-SGM | 99.7 | 99.3 | 91.1 | 13.3 | 7.2 | 2.8 |
| LinBP-MI-DI | 99.5 | 99.2 | 89.3 | 26.1 | 16.5 | 3.2 |
| LinBP-MI-DI-SGM | 99.8 | 99.3 | 90.2 | 32.6 | 22.1 | 4.6 |
| MI-DI+RAP | **99.9** | **100** | **93.7** | **75.1** | **69.7** | **13.9** |

TTP (Naseer et al., 2021) is the state-of-the-art generative method. To compare with it, we adopt the generators based on ResNet-50 provided by the authors. Since TTP needs to train the perturbation generator for each targeted class, we follow their "10-Targets (all-source)" setting, as did in Zhao et al. (2020). The results are shown in Table 6, where our MTDSI+RAP-LS behaves best and outperforms TTP and MTDI by large margins of 14.9% and 25.7%, respectively.

Table 6: The comparison with TTP on targeted attack. The best results are bold.

| Attack | Dense-121 | VGG-16 | Inc-v3 |
|---|---|---|---|
| TTP | 79.6 | 78.6 | 40.3 |
| MTDI | 78.6 | 74.6 | 12.7 |
| MTDI+RAP-LS | 90.8 | 87.2 | 35.4 |
| MTDSI | 93.2 | 80.0 | 41.3 |
| MTDSI+RAP-LS | **95.7** | **88.1** | **59.3** |

Table 4: The **targeted attack success rate** (%) **of combinational methods with RAP**. The results with logit loss are reported. The best results are bold and the second best results are underlined.

| Attack | ResNet-50 $\Longrightarrow$ | | | DenseNet-121 $\Longrightarrow$ | | |
|---|---|---|---|---|---|---|
| | Dense-121 | VGG-16 | Inc-v3 | Res-50 | VGG-16 | Inc-v3 |
| MTDI / +RAP / +RAP-LS | 74.9 / 78.2 / **88.5** | 62.8 / 72.9 / **81.5** | 10.9 / 28.3 / **33.2** | 44.9 / 64.3 / **74.5** | 38.5 / 55.0 / **65.5** | 7.7 / 23.0 / **26.5** |
| MTDSI / +RAP / +RAP-LS | 86.3 / 88.4 / **93.3** | 70.1 / 77.7 / **84.7** | 38.1 / 51.8 / **58.0** | 55.0 / 71.2 / **75.8** | 42.0 / 58.4 / **62.3** | 19.8 / 39.0 / **39.2** |
| MTDAI / +RAP / +RAP-LS | 91.4 / 89.4 / **93.6** | 79.9 / 79.0 / **86.3** | 50.8 / 57.1 / **64.1** | 69.1 / 74.2 / **82.1** | 54.7 / 63.1 / **69.3** | 32.0 / 43.5 / **49.3** |
| Attack | VGG-16 $\Longrightarrow$ | | | Inc-v3 $\Longrightarrow$ | | |
| | Res-50 | Dense-121 | Inc-v3 | Res-50 | Dense-121 | VGG-16 |
| MTDI / +RAP / +RAP-LS | 11.8 / 16.7 / **22.9** | 13.7 / 19.4 / **27.4** | 0.7 / 3.4 / **4.6** | 1.8 / **8.3** / 7.5 | 4.1 / **14.8** / 13.4 | 2.9 / 8.0 / **9.8** |
| MTDSI / +RAP / +RAP-LS | 31.0 / 35.3 / **38.7** | 41.7 / 44.4 / **49.6** | 9.6 / **15.2** / 13.7 | 5.6 / **11.9** / 10.7 | 10.4 / 21.2 / **20.9** | 4.2 / **8.9** / 8.6 |
| MTDAI / +RAP / +RAP-LS | 36.2 / 39.0 / **43.1** | 48.0 / 45.1 / **55.2** | 11.6 / 17.1 / **17.6** | 9.6 / 13.6 / **16.7** | 17.9 / 27.5 / **31.6** | 8.4 / 12.0 / **12.1** |

Table 7: The evaluation on diverse network architectures and defense models.

| Attack | Untarged | | | Targeted | | | Untarged | | Targeted | |
|---|---|---|---|---|---|---|---|---|---|---|
| | IncRes-v2 | NASNet-L | ViT-B/16 | IncRes-v2 | NASNet-L | ViT-B/16 | Inc-v3$_{adv}$ | IncRes-v2$_{ens}$ | Inc-v3$_{adv}$ | IncRes-v2$_{ens}$ |
| MTDI | 83.4 | 89.0 | 27.9 | 14.8 | 32.1 | 0.4 | 68.1 | 50.9 | 0.8 | 0.0 |
| MTDI+RAP-LS | 95.6 | 97.5 | 42.7 | 43.0 | 62.5 | 1.7 | 86.5 | 72.3 | 9.7 | 4.1 |
| MTDSI | 95.7 | 98.0 | 43.0 | 45.5 | 67.9 | 2.6 | 90.0 | 79.6 | 12.7 | 6.7 |
| MTDSI+RAP-LS | 98.6 | 99.7 | 57.4 | 64.0 | 80.4 | 5.3 | 96.5 | 91.5 | 31.0 | 22.0 |
| MTDAI | 97.3 | 98.8 | 45.5 | 58.4 | 75.3 | 3.3 | 92.1 | 82.7 | 17.2 | 12.2 |
| MTDAI+RAP-LS | 99.2 | 99.8 | 60.2 | 70.4 | 82.6 | 7.4 | 96.7 | 91.6 | 34.4 | 26.0 |

## 4.6 THE EVALUATION ON DIVERSE NETWORK ARCHITECTURES AND DEFENSE MODELS

To further demonstrate the efficacy of RAP, we evaluate our method on more diverse network architectures, including Inception-ResNet-v2, NASNet-Large and ViT-Base/16. We adopt ResNet-50 as the surrogate model and the results are shown in Table 7, *col 2-7*. As shown in the table, the proposed RAP-LS achieves significant improvements for all three combinational methods on all target models, and MTDAI+RAP-LS achieves the best performance for diverse models. For MTDAI, the average performance improvements induced by RAP-LS is 5.9% and 7.8% for untargeted and targeted attacks, respectively. Since ViT is based on the transformer architecture that totally being different from convolution models, the transfer attacks based on Resnet-50 behave relatively poor on it, especially on targeted attacks. Yet our RAP-LS still obtains consistent improvements for all compared methods. We also consider the ensemble-model attack on these diverse network architectures and the results are given in *supplementary materials*.

Furthermore, we evaluate RAP on attacking defense models adv-Inc-v3 and ens-adv-Inc-Res-v2. Following prior works (Wang et al., 2021b; Xie et al., 2019b), we adopt the ensemble-model attack by averaging the logits of different surrogate models, including ResNet-50, ResNet-101, Inception-v3, and Inception-ResNet-v2. The transfer attack success rate on defense models are shown in Table 7, *col 8-11*. We can observe that our RAP-LS further boosts transferability of the baseline methods on both targeted and untargeted attacks. For untargeted attacks, RAP-LS achieves average performance improvements of $9.8\%$ and $14.1\%$ on Inc-v3$_{adv}$ and IncRes-v2$_{ens}$, respectively. For targeted attacks, the average performance improvements of RAP-LS are $14.8\%$ and $11.1\%$, respectively.

## 4.7 ABLATION STUDY

We conduct ablation study on the hyper-parameters of the proposed RAP, including the size of neighborhoods $\epsilon_n$, the iteration number of inner optimization $T$ and late-start $K_{LS}$. We adopt targeted attacks with ResNet-50 as the surrogate model.

We first evaluate the effect of $\epsilon_n$ and $T$. We consider different values of $\epsilon_n$, including $2/255$, $4/255$, $8/255$, $12/255$, $16/255$, and $20/255$. In Figure 3 (a), we plot the tendency curves of the targeted attack success rate under different values of $\epsilon_n$ and $T$. Note that in Sec. 3.2, we set $\alpha_n = \epsilon_n/T$. Thus for a fixed $\epsilon_n$, larger $T$ indicating lower stepsize $\alpha_n$. The minimum stepsize of $\alpha_n$ is set to $2/255$. We have the following observation from the plot: for a fixed $\epsilon_n$, the more iterations $T$, the better attack performance. Thus, we adopt a relatively smaller $\alpha_n = 2/255$ in our experiments. In Figure 3 (b-d), we further plot the results of different attack methods and target models *w.r.t.* $\epsilon_n$, where $\alpha_n = 2/255$. As shown in the plots, the larger $\epsilon_n$ generally improves the attack performance. For Inception-v3 and DenseNet-121, the improvements become mild for even larger $\epsilon_n$. Overall, the value of 12 or 16 could lead to satisfactory result under most cases.

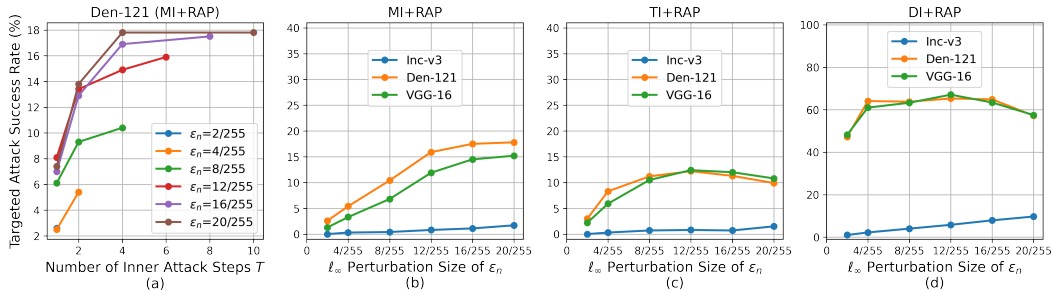

Figure 3: Targeted attack success rate ($\%$) with various $T$ and $\epsilon_n$. Res-50 is set as surrogate model.

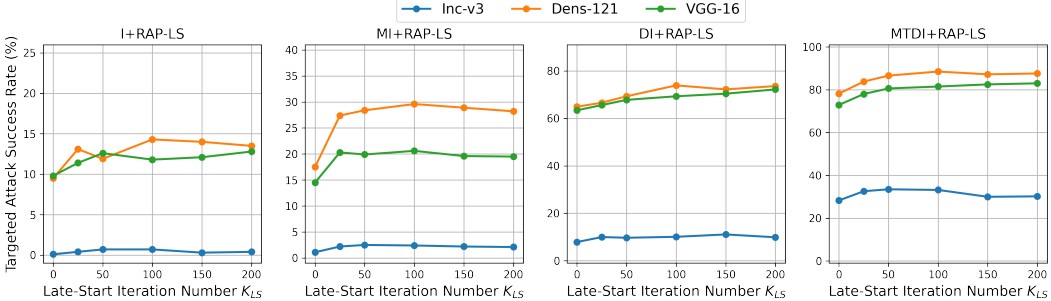

Figure 4: Targeted attack success rate ($\%$) with various $K_{LS}$. Res-50 is set as surrogate model.

Then we conduct the ablation study of $K_{LS}$. In Figure 4, we report the targeted attack success rate of I, MI, DI, and MI-TI-DI combined with RAP-LS with $K_{LS} = 0, 25, 50, 100, 150, 200$. Note that the RAP-LS with $K_{LS} = 0$ reduces to RAP. As shown in the plots, the proposed late-start strategy can further boost attack performance of RAP for most cases. In general, the performance improvements increase as $K_{LS}$ increases, and then become mild when $K_{LS}$ is larger than 100. The suitable value of $K_{LS}$ is relatively consistent among different methods and target models.

## 4.8 THE TARGETED ATTACK AGAINST GOOGLE CLOUD VISION API

Finally, we conduct the transfer attacks to attack a practical and widely used image recognition system, Google Cloud Vision API, and in the more challenging targeted attack scenario. MTDAI-RAP-LS behaves the best performance in above experiments, so we choose it to conduct the attack. We take the evaluation on randomly selected 500 images and use ResNet-50 as surrogate model. As the API returns 10 predicted labels for each query, to evaluate the attacking performance, we test whether or not the target class appears in the returned predictions. Since the predicted label space of Google Cloud Vision API do not fully correspond to the 1000 ImageNet classes, we manually treat classes with similar semantics to be the same classes. In comparison, the baseline MTDAI successfully attacks 232 images against the Google API. Our RAP-LS achieves a large improvement, successfully attacking 342 images, leading to a 22.0% performance improvements. These demonstrates the high efficacy of our method to improve transferability on real-world system.

## 5 CONCLUSION

In this work, we study the transferability of adversarial examples that is significant for black-box attacks. The transferability of adversarial examples is generally influenced by the overfitting of surrogate models. To alleviate this, we propose to seeking adversarial examples that locate at flatter local regions. That is, instead of optimizing the pinpoint adversarial loss, we aim to obtain a consistently low loss at the neighbor regions of the adversarial examples. We formulate this as a min-max bi-level optimization problem, where the inner maximization aims to inject the worse-case perturbation for the adversarial examples. We conduct a rigorous experimental study, covering untargeted attack and targeted attack, standard and defense models, and a real-world Google Cloud Vision API. The experimental results demonstrate that RAP can significantly boost the transferability of adversarial examples.

ETHICS STATEMENT

Deep neural networks (DNNs) have been successfully applied in many safety-critical tasks, such as autonomous driving, face recognition and verification, *etc*. And adversarial samples have posed a serious threat to machine learning systems. For real-world applications, the DNN model as well as the training dataset, are often hidden from users. Therefore, the attackers need to generate the adversarial examples under black-box setting where they do not know any information of the target model. For black-box setting, the adversarial transferability matters since it can allow the attackers to attack target models by using adversarial examples generated on the surrogate models. This work can potentially contribute to understanding of transferability of adversarial examples. Besides, the better transferability of adversarial examples calls the machine learning and security communities into action to create stronger defenses and robust models against black-box attacks.

REPRODUCIBILITY STATEMENT

Our approach is easy to implement and requires only simple modifications to the existing methods, as shown in Sec.3.2. The detailed algorithm of our method is shown in Algorithm 1. The implementation details of our method are shown in Sec.4.1. We provide the details to reproduce the experimental results in Sec.4.1 of main submission and Sec.A of supplementary materials, including datasets in Sec.4.1, evaluated models in Sec.4.1, baseline methods in Sec.4.1. The implementation details of baseline methods also shown in Sec.A. We will release all related codes after the acceptance of this work.

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

# A  IMPLEMENTATION DETAILS

**Implementation Details of Evaluated Models.**   For ResNet-50, DenseNet-121, VGG-16, Inception-v3, we adopt the pre-trained models provided by torchvision package. For Inception-ResNet-v2, NASNet-Large, ViT-Base/16, adv-Inc-v3, and ens-adv-Inc-Res-v2, we adopt the provided pre-trained models[2].

**Implementation Details of Baseline Attack Methods.**   We adopt the source code [3] provided by Zhao et al. (2020) to implement I, MI, TI, and DI attacks. The decay factor for MI is set as $1.0$. The kernel size is set as $5$ for TI attack, following Gao et al. (2020). The transformation probability is set as $0.7$ for DI. For SI and Admix, we adopt the parameters suggested in Wang et al. (2021b). The number of copies for SI is set as $5$. The number of randomly sample $m_2$ and $\eta$ of Admix are set as $3$ and $0.2$ respectively. For implementation of ILA and LinBP, we utilize the source code [4] provided by Guo et al. (2020). For implementation of TTP, we use the pre-trained generator [5] based on ResNet-50 provided by Naseer et al. (2021).

**Implementation Details of Visualization.**   We visualize the flatness of the loss landscape around $x^{adv}$ on surrogate model by plotting the loss change when moving $x^{adv}$ along a random direction with different magnitudes. Specially, we first sample $d$ from a Gaussian distribution and normalize it on a $\ell_2$ unit norm ball, $d \leftarrow \frac{d}{\|d\|_F}$. Then, we calculate the loss change (flatness) $f(a)$ with different magnitudes $a$,

$$f(a) = \mathcal{L}(\mathcal{M}^s(\mathcal{G}(x^{adv} + a \cdot d); \theta), y_t) - \mathcal{L}(\mathcal{M}^s(\mathcal{G}(x^{adv}); \theta), y_t). \tag{6}$$

Considering $d$ is randomly selected, we repeat the above calculation 20 times with different $d$ and take the averaged value to conduct the visualization.

**Computation Cost.**   We conducted all experiments in an Nvidia-V100 GPU. Taking MTDI as an example, its computation time is 2 hours. Combining RAP or RAP-LS, the computation time becomes 5.5 or 4 hours.

# B  THE EXPERIMENTAL RESULTS IN DISCUSSION

In this section, we show the experiments mentioned in the discussion. The detailed results are shown in the below tables and figures.

**Visualization of Untargeted Attacks**   We add the visualization results about untargeted attacks. The implementation details are also shown in Section A.

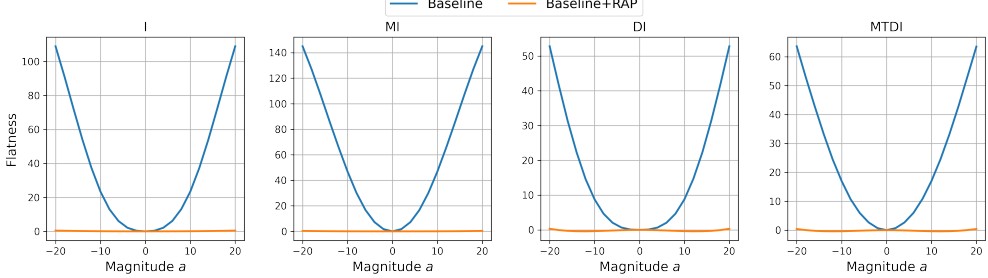

Figure 5: The flatness visualization of untargeted adversarial examples.

We add I-FGSM with random-start (I*) in our experiments. The untargeted attack performance of I* and I* combined with RAP is shown in Table 8. We follow the experimental setting in Section 4.3 of the main submission. RAP achieves improvements for I* on each target model, and with late-start, RAP-LS further enhances the transfer attack performance for I*, getting a $6.0\%$ increase in terms of average attack success rate.

---

[2] https://github.com/rwightman/pytorch-image-models
[3] https://github.com/ZhengyuZhao/Targeted-Tansfer
[4] https://github.com/qizhangli/linbp-attack
[5] https://github.com/Muzammal-Naseer/TTP

Table 8: The **untargeted attack success rate** (%) **of I\* with RAP**. The results with $CE$ loss and 400 iterations are reported. The best results are bold and the second best results are underlined.

| Attack | ResNet-50 $\Longrightarrow$ | | | DenseNet-121 $\Longrightarrow$ | | |
|---|---|---|---|---|---|---|
| | Dense-121 | VGG-16 | Inc-v3 | Res-50 | VGG-16 | Inc-v3 |
| I\* / +RAP / +RAP-LS | 75.7 / 80.2 / **81.2** | 78.0 / 80.5 / **82.1** | 35.0 / 47.2 / **48.9** | 85.6 / 86.1 / **86.9** | 83.7 / 84.6 / **85.0** | 47.0 / 54.2 / **55.1** |
| Attack | VGG-16 $\Longrightarrow$ | | | Inc-v3 $\Longrightarrow$ | | |
| | Res-50 | Dense-121 | Inc-v3 | Res-50 | Dense-121 | VGG-16 |
| I\* / +RAP / +RAP-LS | 52.9 / 56.8 / **57.2** | 49.9 / 54.5 / **55.3** | 21.7 / 24.5 / **25.1** | 50.9 / 55.2 / **57.9** | 48.6 / 54.0 / **56.2** | 54.7 / 61.6 / **63.8** |

**The Evaluation on More Defense Models** Following the reviewers' suggestions, we take the evaluations on more defense models: Feature Denoising (Xie et al., 2019a), Adversarial training models in ImageNet (Salman et al., 2020), NRP (Naseer et al., 2020), R&P (Xie et al., 2018).

For Feature Denoising, we utilize the pre-trained ResNet-152 model. For AT models on ImageNet, we adopt the pre-trained ResNet-50 AT models from (Salman et al., 2020). For $\ell_\infty$ norm, we adopt the ResNet-50 AT model with budget $4/255$, which ranks first in the RobustBench leaderboard [6]. For $\ell_2$ norm, we adopt the ResNet-50 AT model with budget $0.5$. The untargeted attack performance is shown in Table 9. We follow the experimental settings in Section 4.6 of the main submission. We can observe that our RAP-LS further boosts the transferability of baseline methods on these new defense models, getting a $5.5\%$ boost for the average attack success rate.

For NRP, we adopt the pre-trained purifiers provided by the authors. Since NRP is an offline defense module, we combine it with the two used ensemble AT models and the above two AT models. The untargeted attack performance is shown in Table 10. We also follow the experimental settings in Section 4.6 of the main submission. Combining NRP with AT models is a much stronger defense mechanism, but RAP-LS still achieves an improvement by $0.8\%$.

For R&P, we adopt the source code provided by Dong et al. (2020) to implement it. We also combine R&P with the two used ensemble AT models and the two new AT models above. The untargeted attack performance is shown in Table 11. We also follow the experimental settings in Section 4.6 of the main submission. For R&P, RAP-LS achieves an $9.1\%$ increase in terms of average attack success rate.

Table 9: The evaluation of ensemble attacks on **two AT models** and **Feature Noising**.

| Attack | Untarged | | |
|---|---|---|---|
| | Res-50 AT ($\ell_2$) | Res-50 AT ($\ell_\infty$) | Feature Denoising |
| MTDI | 42.5 | 32.4 | 44.1 |
| MTDI+RAP-LS | **59.5** | **34.4** | **44.4** |
| MTDSI | 56.6 | 35.8 | 45.0 |
| MTDSI+RAP-LS | **70.3** | **36.6** | **45.7** |
| MTDAI | 62.1 | 35.6 | 44.2 |
| MTDAI+RAP-LS | **73.7** | **37.7** | **45.2** |

Table 10: The evaluation of ensemble attacks on defense models with **NRP**.

| Attack | Untarged | | | |
|---|---|---|---|---|
| | Inc-v3$_{adv}$ | IncRes-v2$_{ens}$ | Res-50 AT ($\ell_2$) | Res-50 AT ($\ell_\infty$) |
| MTDI | **23.1** | 13.5 | 14.2 | 25.7 |
| MTDI+RAP-LS | 22.7 | **14.8** | **14.9** | **26.3** |
| MTDSI | 22.5 | 14.2 | 15.0 | 26.1 |
| MTDSI+RAP-LS | **24.5** | **15.3** | **15.4** | **26.2** |
| MTDAI | 24.1 | 14.7 | 14.2 | 25.9 |
| MTDAI+RAP-LS | **24.9** | **15.6** | **15.3** | **26.1** |

**The Comparison with EOT Attack** We sample noise $n^{adv}$ from Gaussian multiple times and add them to $x^{adv}$ in each iteration. We set the standard deviation of Gaussian noise as $15/255$ and the

---

[6]https://robustbench.github.io

Table 11: The evaluation of ensemble attacks on defense models with **R&P**.

| Attack | Untarged | | | |
|---|---|---|---|---|
| | Inc-v3$_{adv}$ | IncRes-v2$_{ens}$ | Res-50 AT ($\ell_2$) | Res-50 AT ($\ell_\infty$) |
| MTDI | 65.0 | 46.2 | 52.5 | 43.7 |
| MTDI+RAP-LS | **82.1** | **63.2** | **65.3** | **45.8** |
| MTDSI | 86.5 | 69.6 | 64.1 | 45.9 |
| MTDSI+RAP-LS | **93.4** | **84.9** | **74.0** | **46.2** |
| MTDAI | 88.9 | 76.5 | 68.4 | 46.2 |
| MTDAI+RAP-LS | **94.8** | **87.0** | **77.7** | **47.7** |

number of EOT as 20. We take MTDI as our baseline, ResNet and DenseNet as surrogate models and compare this attack with our method RAP.

The untargeted and targeted attack performance is shown in Table 12. As shown in experimental results, our methods achieve better performance and surpass this EOT attack a large margin on all target models for targeted attacks. RAP-LS gets an 8.7% increase for targeted attacks in terms of average success rate. This demonstrates the superiority of our methods.

Table 12: The comparison between **EOT** and our method.

| Attack | ResNet-50 (Untarged) $\implies$ | | | ResNet-50 (Ttarged) $\implies$ | | |
|---|---|---|---|---|---|---|
| | Dense-121 | VGG-16 | Inc-v3 | Dense-121 | VGG-16 | Inc-v3 |
| MTDI+EOT / MTDI+RAP-LS | 99.9 / **100** | 99.7 / **99.9** | 95.3 / **96.9** | 82.7 / **88.5** | 73.4 / **81.5** | **34.6** / 33.2 |

| Attack | DenseNet-121 (Untarged) $\implies$ | | | DenseNet-121 (Ttarged) $\implies$ | | |
|---|---|---|---|---|---|---|
| | Res-50 | VGG-16 | Inc-v3 | Res-50 | VGG-16 | Inc-v3 |
| MTDI+EOT / MTDI+RAP-LS | 99.8 / **100** | 99.7 / **100** | 96.3 / **97.1** | 56.6 / **74.5** | 49.7 / **65.5** | 22.6 / **26.5** |

## C  ADDITIONAL EXPERIMENTAL RESULTS

In this section, we first show the evaluation of targeted attacks with CE loss in Sec.C.1. Then we show the results of ensemble attacks on more diverse network architectures in Sec.C.2. In Sec.C.3, we report the experimental results *w.r.t.* different value of iterations.

### C.1  THE RESULTS OF TARGETED ATTACKS WITH CE LOSS

Following the settings in main submission, we evaluate the targeted attack performance of the different baseline methods with our method on ResNet-50, DenseNet-121, VGG-16, and Inception-v3. The results of combinational methods are shown in Table 13. The RAP-LS outperforms all combinational methods by a significantly margin. Taking the average attack success rate of all target models as the evaluation metric, RAP-LS achieves 20.9%, 18.4%, and 15.1% improvements over the MTDI, MTDSI and MTDAI, respectively.

Table 13: The **targeted attack success rate (%) of combinational methods with RAP**. The results with $CE$ loss and 400 iterations are reported. The best results are bold and the second best results are underlined.

| Attack | ResNet-50 $\implies$ | | | DenseNet-121 $\implies$ | | |
|---|---|---|---|---|---|---|
| | Dense-121 | VGG-16 | Inc-v3 | Res-50 | VGG-16 | Inc-v3 |
| MTDI / +RAP / +RAP-LS | 45.5 / 78.3 / **85.9** | 29.8 / 70.5 / **76.7** | 4.5 / 21.3 / **25.3** | 20.0 / 54.0 / **62.7** | 9.9 / 41.7 / **48.7** | 2.6 / 17.5 / **18.5** |
| MTDSI / +RAP / +RAP-LS | 77.7 / 89.0 / **93.7** | 39.9 / 69.4 / **76.7** | 26.9 / 45.3 / **50.8** | 30.5 / 60.4 / **69.5** | 14.9 / 42.8 / **49.7** | 12.7 / 26.6 / **32.5** |
| MTDAI / +RAP / +RAP-LS | 90.2 / 91.4 / **96.1** | 61.8 / 73.7 / **83.4** | 44.5 / 47.9 / **59.0** | 55.8 / 68.4 / **79.3** | 35.1 / 51.8 / **64.1** | 26.3 / 32.4 / **40.4** |

| Attack | VGG-16 $\implies$ | | | Inc-v3 $\implies$ | | |
|---|---|---|---|---|---|---|
| | Res-50 | Dense-121 | Inc-v3 | Res-50 | Dense-121 | VGG-16 |
| MTDI / +RAP / +RAP-LS | 0.5 / 10.4 / **12.1** | 0.1 / 11.0 / **13.5** | 0.0 / 1.7 / **2.0** | 2.2 / 4.9 / **5.9** | 2.2 / 9.8 / **11.0** | 1.2 / 4.9 / **6.7** |
| MTDSI / +RAP / +RAP-LS | 5.4 / 17.4 / **16.8** | 9.5 / **28.4** / 25.2 | 2.2 / **7.1** / 5.1 | 4.4 / 8.6 / **8.9** | 7.9 / 16.3 / **19.3** | 2.0 / **6.4** / **6.4** |
| MTDAI / +RAP / +RAP-LS | 11.6 / 22.6 / **26.6** | 20.6 / 32.1 / **39.1** | 5.1 / 9.2 / **9.5** | 6.7 / 12.3 / **17.0** | 14.0 / 22.9 / **29.2** | 4.5 / 9.4 / **13.2** |

## C.2 The Results of Ensemble Attacks on Diverse Network Architectures

We also take the evaluation of the ensemble attacks on diverse network architecture (Sec.4.6). We adopt the ensemble-model attack by averaging the logits of different surrogate models, including ResNet-50, DenseNet-121, VGG-16, and Inception-v3. The transfer attack success rate on diverse models are shown in Table 14. Compared with results of single model attack in Table 7, the ensemble attack achieve the better performance. We can observe that our RAP-LS further boosts transferability of the baseline methods on both targeted and untargeted attacks. We take ViT as target model for example. For untargeted attacks, RAP-LS achieves average performance improvements of 19.2%. For targeted attacks, RAP-LS achieves average performance improvements of 10.4%.

Table 14: The evaluation of ensemble attacks on diverse network architectures.

| Attack | Untarged | | | Targeted | | |
|---|---|---|---|---|---|---|
| | IncRes-v2 | NASNet-L | ViT-B/16 | IncRes-v2 | NASNet-L | ViT-B/16 |
| MTDI | 98.6 | 99.3 | 46.2 | 65.7 | 80.1 | 2.8 |
| MTDI+RAP-LS | **100** | **100** | **73.2** | **84.4** | **89.7** | **12.7** |
| MTDSI | 99.8 | 100 | 68.3 | 81.7 | 89.4 | 15.0 |
| MTDSI+RAP-LS | **100** | **100** | **85.0** | **89.8** | **92.3** | **25.1** |
| MTDAI | 100 | 100 | 70.7 | 88.8 | 91.2 | 16.8 |
| MTDAI+RAP-LS | **100** | **100** | **84.6** | **90.4** | **91.8** | **27.8** |

## C.3 The Experimental Results *w.r.t.* Different Value of Iterations

In the main submission, we report the evaluations of $K = 400$. Here, we further report the performance with different values of $K$ for completeness in Table 15 (targeted attack) and Table 16 (untargeted attack). From the results, we observe that the attacking performance generally increase as $K$ increases for most cases, this is also aligned with prior works (Zhao et al., 2020).

Table 15: The targeted attack success rate (%) of all baseline attacks with our method. The results with logit loss and 10/100/200/300/400 iterations are reported. We highlight the results with $K = 400$ in bold.

| | ResNet-50 → Inception-v3 | | |
|---|---|---|---|
| | Baseline | +RAP | +RAP-LS |
| I | 0.0 / 0.1 / 0.2 / 0.1 / **0.1** | 0.0 / 0.2 / 0.3 / 0.3 / **0.1** | 0.0 / 0.1 / 0.4 / 0.6 / **0.7** |
| MI | 0.1 / 0.1 / 0.2 / 0.1 / **0.1** | 0.0 / 0.6 / 1.0 / 1.0 / **1.1** | 0.1 / 0.1 / 1.4 / 1.6 / **2.4** |
| TI | 0.0 / 0.3 / 0.2 / 0.2 / **0.1** | 0.0 / 0.7 / 0.9 / 1.2 / **0.8** | 0.0 / 0.3 / 1.3 / 1.3 / **1.2** |
| DI | 0.2 / 1.2 / 1.7 / 1.5 / **1.5** | 0.0 / 3.8 / 6.6 / 7.7 / **7.9** | 0.2 / 1.2 / 10.2 / 9.4 / **10.1** |
| SI | 0.3 / 2.6 / 2.4 / 2.0 / **1.8** | 0.2 / 6.6 / 8.2 / 8.6 / **9.3** | 0.3 / 2.6 / 9.6 / 9.3 / **10.5** |
| Admix | 1.4 / 5.7 / 5.9 / 6.0 / **5.8** | 0.6 / 14.6 / 16.6 / 16.5 / **17.1** | 1.4 / 5.7 / 18.5 / 19.2 / **19.6** |
| MI-TI-DI | 1.5 / 7.9 / 9.8 / 10.5 / **10.9** | 0.1 / 12.7 / 22.3 / 26.3 / **28.3** | 1.5 / 7.9 / 26.8 / 30.0 / **33.2** |
| MI-TI-DI-SI | 8.9 / 34.1 / 36.7 / 38.1 / **38.1** | 3.3 / 43.3 / 47.9 / 49.9 / **51.8** | 8.9 / 34.8 / 54.8 / 55.8 / **58.0** |
| MI-TI-DI-Admix | 13.5 / 45.7 / 49.2 / 50.5 / **50.8** | 5.0 / 48.1 / 53.4 / 56.2 / **57.1** | 13.5 / 45.1 / 61.4 / 63.0 / **64.1** |

| | ResNet-50 → DenseNet-121 | | |
|---|---|---|---|
| | Baseline | +RAP | +RAP-LS |
| I | 0.9 / 5.3 / 5.0 / 5.5 / **4.5** | 0.0 / 4.8 / 7.9 / 8.8 / **9.5** | 0.9 / 5.3 / 14.0 / 14.0 / **14.3** |
| MI | 3.4 / 6.3 / 6.3 / 6.0 / **6.3** | 0.2 / 9.0 / 14.1 / 15.8 / **17.5** | 3.4 / 6.3 / 25.9 / 28.9 / **29.6** |
| TI | 2.5 / 8.6 / 8.9 / 9.0 / **7.2** | 0.0 / 7.1 / 10.1 / 11.2 / **11.0** | 2.5 / 8.6 / 16.1 / 16.4 / **17.3** |
| DI | 8.4 / 54.8 / 60.4 / 61.2 / **62.6** | 0.1 / 40.6 / 53.2 / 59.4 / **64.9** | 8.4 / 54.6 / 70.9 / 72.5 / **73.9** |
| SI | 9.7 / 29.6 / 30.4 / 30.4 / **30.0** | 2.2 / 45.8 / 50.9 / 52.5 / **53.2** | 9.7 / 29.6 / 60.0 / 61.1 / **61.1** |
| Admix | 23.6 / 55.6 / 55.5 / 55.6 / **54.6** | 5.3 / 61.2 / 66.0 / 66.9 / **68.0** | 23.6 / 55.6 / 74.4 / 74.7 / **74.6** |
| MI-TI-DI | 16.3 / 66.9 / 71.4 / 73.4 / **74.9** | 1.8 / 56.7 / 71.2 / 76.4 / **78.2** | 16.3 / 66.7 / 85.2 / 85.7 / **88.5** |
| MI-TI-DI-SI | 41.0 / 82.8 / 84.5 / 86.2 / **86.3** | 12.9 / 80.2 / 85.7 / 87.8 / **88.4** | 41.0 / 82.5 / 91.9 / 92.4 / **93.3** |
| MI-TI-DI-Admix | 48.0 / 88.7 / 90.9 / 91.1 / **91.4** | 20.3 / 83.2 / 87.2 / 88.4 / **89.4** | 47.9 / 88.5 / 93.5 / 93.8 / **93.6** |

| | ResNet-50 → VGG-16 | | |
| | Baseline | +RAP | +RAP-LS |
|---|---|---|---|
| I | 1.0 / 2.7 / 2.6 / 2.3 / **2.4** | 0.0 / 5.6 / 8.3 / 9.8 / **9.8** | 1.0 / 2.7 / 11.4 / 12.8 / **11.8** |
| MI | 1.2 / 2.1 / 2.4 / 2.2 / **2.2** | 0.1 / 8.6 / 12.3 / 14.1 / **14.5** | 1.2 / 2.1 / 18.2 / 20.0 / **20.6** |
| TI | 1.1 / 4.8 / 4.8 / 4.5 / **4.0** | 0.1 / 6.0 / 9.3 / 9.9 / **12.9** | 1.1 / 4.8 / 14.2 / 15.3 / **15.3** |
| DI | 7.6 / 51.0 / 56.9 / 56.6 / **57.2** | 0.4 / 42.4 / 55.0 / 61.5 / **63.4** | 7.6 / 51.0 / 69.3 / 69.8 / **69.3** |
| SI | 4.4 / 10.4 / 8.9 / 8.8 / **9.5** | 1.1 / 27.8 / 31.1 / 30.7 / **32.8** | 4.4 / 10.4 / 35.8 / 35.1 / **36.0** |
| Admix | 10.6 / 24.9 / 25.0 / 26.2 / **26.0** | 3.6 / 41.4 / 45.2 / 43.8 / **45.4** | 10.6 / 24.9 / 51.7 / 51.9 / **51.6** |
| MI-TI-DI | 12.1 / 55.9 / 61.0 / 63.9 / **62.8** | 1.5 / 53.0 / 64.7 / 70.9 / **72.9** | 12.1 / 55.8 / 78.5 / 81.7 / **81.5** |
| MI-TI-DI-SI | 24.6 / 67.4 / 68.5 / 69.7 / **70.1** | 8.2 / 66.4 / 73.7 / 75.2 / **77.7** | 24.5 / 66.4 / 82.4 / 83.7 / **84.7** |
| MI-TI-DI-Admix | 33.4 / 75.3 / 77.5 / 78.7 / **79.9** | 14.6 / 70.4 / 76.7 / 78.3 / **79.0** | 33.3 / 75.2 / 85.4 / 86.4 / **86.3** |

| | DenseNet121 → Inception-v3 | | |
| | Baseline | +RAP | +RAP-LS |
|---|---|---|---|
| I | 0.0 / 0.1 / 0.2 / 0.1 / **0.0** | 0.0 / 0.6 / 0.9 / 0.7 / **0.8** | 0.0 / 0.1 / 1.0 / 1.3 / **1.2** |
| MI | 0.2 / 0.2 / 0.3 / 0.3 / **0.3** | 0.0 / 1.2 / 2.1 / 2.1 / **2.0** | 0.2 / 0.2 / 2.5 / 3.7 / **3.4** |
| TI | 0.0 / 0.4 / 0.3 / 0.5 / **0.2** | 0.0 / 1.2 / 1.5 / 1.6 / **2.1** | 0.0 / 0.4 / 2.6 / 3.1 / **3.0** |
| DI | 0.3 / 1.9 / 1.4 / 1.7 / **1.4** | 0.0 / 4.1 / 7.0 / 7.6 / **8.8** | 0.3 / 1.9 / 9.3 / 9.9 / **10.0** |
| SI | 0.3 / 1.5 / 1.8 / 1.6 / **1.6** | 0.1 / 7.6 / 9.2 / 10.0 / **8.5** | 0.3 / 1.5 / 9.2 / 10.7 / **10.4** |
| Admix | 1.7 / 5.0 / 5.4 / 5.5 / **5.0** | 0.2 / 15.8 / 17.0 / 17.7 / **17.1** | 1.7 / 5.0 / 18.5 / 18.2 / **17.6** |
| MI-TI-DI | 1.2 / 6.8 / 7.9 / 8.7 / **7.7** | 0.1 / 13.0 / 19.7 / 22.2 / **23.0** | 1.2 / 6.7 / 21.9 / 26.2 / **26.5** |
| MI-TI-DI-SI | 5.1 / 17.6 / 18.9 / 19.3 / **19.8** | 2.0 / 30.4 / 35.1 / 37.0 / **39.0** | 5.2 / 17.7 / 36.8 / 38.9 / **39.2** |
| MI-TI-DI-Admix | 11.4 / 30.5 / 32.2 / 31.4 / **32.0** | 3.9 / 36.7 / 41.3 / 42.2 / **43.5** | 11.2 / 31.2 / 47.2 / 49.2 / **49.3** |

| | DenseNet121 → ResNet-50 | | |
| | Baseline | +RAP | +RAP-LS |
|---|---|---|---|
| I | 1.8 / 6.5 / 5.6 / 5.5 / **5.0** | 0.2 / 7.7 / 11.2 / 12.4 / **12.8** | 1.8 / 6.5 / 18.7 / 19.0 / **17.9** |
| MI | 3.4 / 5.4 / 5.2 / 4.9 / **4.6** | 0.3 / 10.2 / 14.3 / 16.3 / **16.2** | 3.4 / 5.4 / 23.6 / 26.3 / **26.5** |
| TI | 2.6 / 8.1 / 7.9 / 8.4 / **8.4** | 0.2 / 7.8 / 10.9 / 12.1 / **13.5** | 2.6 / 8.1 / 19.2 / 20.2 / **20.8** |
| DI | 6.3 / 30.4 / 33.1 / 32.0 / **30.2** | 0.4 / 33.6 / 44.1 / 48.7 / **52.6** | 6.3 / 30.8 / 58.8 / 60.4 / **60.4** |
| SI | 7.3 / 16.5 / 15.9 / 14.8 / **14.2** | 1.5 / 33.8 / 39.5 / 41.4 / **41.5** | 7.3 / 16.5 / 44.7 / 44.8 / **43.4** |
| Admix | 16.4 / 32.6 / 30.3 / 28.8 / **29.3** | 3.7 / 48.3 / 52.9 / 53.4 / **53.0** | 16.4 / 32.6 / 60.1 / 58.8 / **58.2** |
| MI-TI-DI | 8.3 / 40.3 / 44.6 / 46.3 / **44.9** | 0.9 / 42.0 / 56.4 / 62.4 / **64.3** | 8.3 / 40.1 / 69.5 / 72.8 / **74.5** |
| MI-TI-DI-SI | 18.6 / 52.3 / 54.1 / 56.2 / **55.0** | 6.6 / 60.3 / 67.5 / 70.6 / **71.2** | 18.6 / 52.5 / 73.8 / 75.5 / **75.8** |
| MI-TI-DI-Admix | 27.6 / 66.3 / 69.7 / 69.8 / **69.1** | 12.1 / 66.4 / 70.8 / 73.2 / **74.2** | 27.6 / 66.4 / 81.4 / 82.0 / **82.1** |

| | DenseNet121 → VGG-16 | | |
| | Baseline | +RAP | +RAP-LS |
|---|---|---|---|
| I | 0.6 / 3.8 / 3.5 / 3.5 / **2.9** | 0.1 / 6.2 / 9.3 / 10.5 / **10.1** | 0.6 / 3.8 / 14.5 / 15.7 / **15.9** |
| MI | 1.6 / 2.4 / 2.6 / 2.7 / **3.1** | 0.2 / 8.6 / 12.2 / 13.0 / **13.4** | 1.6 / 2.4 / 19.5 / 21.7 / **23.2** |
| TI | 1.1 / 5.6 / 5.8 / 4.8 / **5.2** | 0.1 / 6.3 / 9.1 / 11.0 / **12.4** | 1.1 / 5.6 / 16.5 / 17.0 / **16.4** |
| DI | 4.1 / 29.8 / 32.7 / 33.1 / **32.1** | 0.2 / 31.5 / 44.7 / 48.7 / **49.5** | 4.1 / 29.9 / 57.2 / 56.5 / **58.9** |
| SI | 2.8 / 9.8 / 8.8 / 8.5 / **8.4** | 0.6 / 25.8 / 28.2 / 31.4 / **31.0** | 2.8 / 9.8 / 33.5 / 35.3 / **35.2** |
| Admix | 10.2 / 23.3 / 22.1 / 21.3 / **21.5** | 1.7 / 39.4 / 42.2 / 43.0 / **42.7** | 10.2 / 23.3 / 49.7 / 49.3 / **48.2** |
| MI-TI-DI | 6.1 / 32.4 / 36.3 / 39.0 / **38.5** | 0.7 / 36.2 / 49.9 / 53.2 / **55.0** | 6.1 / 32.6 / 61.8 / 64.3 / **65.5** |
| MI-TI-DI-SI | 12.4 / 40.2 / 41.9 / 42.2 / **42.0** | 4.6 / 46.9 / 54.0 / 57.0 / **58.4** | 12.4 / 40.0 / 61.3 / 62.4 / **62.3** |
| MI-TI-DI-Admix | 20.0 / 53.2 / 55.0 / 55.7 / **54.7** | 9.4 / 54.8 / 60.1 / 61.8 / **63.1** | 19.9 / 53.1 / 68.1 / 69.7 / **69.3** |

| | VGG-16 → Inception-v3 | | |
| | Baseline | +RAP | +RAP-LS |
|---|---|---|---|
| I | 0.0 / 0.0 / 0.0 / 0.0 / **0.0** | 0.0 / 0.1 / 0.0 / 0.1 / **0.1** | 0.0 / 0.0 / 0.2 / 0.0 / **0.2** |
| MI | 0.0 / 0.0 / 0.0 / 0.0 / **0.0** | 0.0 / 0.0 / 0.2 / 0.0 / **0.0** | 0.0 / 0.0 / 0.2 / 0.5 / **0.3** |
| TI | 0.0 / 0.0 / 0.0 / 0.1 / **0.0** | 0.0 / 0.1 / 0.1 / 0.1 / **0.1** | 0.0 / 0.0 / 0.4 / 0.4 / **0.4** |
| DI | 0.0 / 0.0 / 0.0 / 0.0 / **0.0** | 0.0 / 0.0 / 0.4 / 0.6 / **0.4** | 0.0 / 0.0 / 0.7 / 0.7 / **1.1** |
| SI | 0.0 / 0.4 / 0.3 / 0.2 / **0.2** | 0.0 / 2.0 / 1.5 / 2.0 / **1.7** | 0.0 / 0.6 / 1.6 / 1.9 / **1.8** |
| Admix | 0.1 / 0.7 / 0.8 / 0.6 / **0.7** | 0.0 / 2.7 / 2.2 / 2.3 / **2.4** | 0.1 / 0.0 / 2.3 / 3.0 / **2.8** |
| MI-TI-DI | 0.1 / 1.0 / 0.8 / 1.1 / **0.7** | 0.0 / 1.8 / 2.8 / 3.0 / **3.4** | 0.1 / 0.9 / 3.4 / 4.0 / **4.6** |
| MI-TI-DI-SI | 1.7 / 7.7 / 9.1 / 9.8 / **9.6** | 0.6 / 12.2 / 14.5 / 13.8 / **15.2** | 1.7 / 8.6 / 11.4 / 12.1 / **13.7** |
| MI-TI-DI-Admix | 3.6 / 12.4 / 12.2 / 11.5 / **11.6** | 1.1 / 14.5 / 16.1 / 15.9 / **17.1** | 3.4 / 11.2 / 15.9 / 17.4 / **17.6** |

| | VGG-16 → ResNet-50 | | |
|---|---|---|---|
| | Baseline | +RAP | +RAP-LS |
| I | 0.2 / 0.4 / 0.3 / 0.3 / **0.1** | 0.0 / 1.0 / 0.8 / 0.8 / **0.7** | 0.2 / 0.5 / 1.4 / 1.5 / **1.4** |
| MI | 0.4 / 0.5 / 0.6 / 0.5 / **0.5** | 0.2 / 1.1 / 1.3 / 1.3 / **1.3** | 0.4 / 0.2 / 2.1 / 2.4 / **1.9** |
| TI | 0.3 / 1.0 / 0.7 / 0.9 / **0.7** | 0.0 / 1.4 / 1.5 / 1.4 / **1.2** | 0.3 / 1.0 / 3.0 / 3.3 / **3.2** |
| DI | 0.5 / 2.8 / 3.1 / 3.4 / **2.8** | 0.0 / 4.9 / 6.7 / 6.5 / **7.3** | 0.5 / 3.9 / 9.5 / 10.1 / **9.7** |
| SI | 1.4 / 4.4 / 3.9 / 3.8 / **3.3** | 0.4 / 9.2 / 9.0 / 9.1 / **9.8** | 1.4 / 4.3 / 10.1 / 9.4 / **9.8** |
| Admix | 4.6 / 7.3 / 6.7 / 5.8 / **5.6** | 0.7 / 10.6 / 11.3 / 10.9 / **11.1** | 4.7 / 7.3 / 11.6 / 12.5 / **11.9** |
| MI-TI-DI | 1.8 / 10.2 / 11.7 / 11.9 / **11.8** | 0.0 / 10.8 / 14.6 / 15.7 / **16.7** | 1.8 / 9.5 / 20.2 / 21.6 / **22.9** |
| MI-TI-DI-SI | 8.8 / 30.1 / 31.6 / 30.3 / **31.0** | 3.2 / 30.8 / 32.5 / 33.5 / **35.3** | 9.0 / 29.5 / 36.9 / 38.5 / **38.7** |
| MI-TI-DI-Admix | 15.2 / 34.6 / 35.1 / 36.6 / **36.2** | 5.4 / 34.7 / 37.3 / 38.1 / **39.0** | 15.3 / 35.5 / 43.2 / 42.9 / **43.1** |

| | VGG-16 → DenseNet-121 | | |
|---|---|---|---|
| | Baseline | +RAP | +RAP-LS |
| I | 0.1 / 0.2 / 0.4 / 0.3 / **0.2** | 0.0 / 0.7 / 1.1 / 0.7 / **1.4** | 0.1 / 0.3 / 1.2 / 1.5 / **1.7** |
| MI | 0.3 / 0.8 / 0.6 / 0.6 / **0.5** | 0.0 / 1.1 / 1.4 / 2.1 / **2.3** | 0.3 / 0.6 / 2.4 / 3.2 / **3.0** |
| TI | 0.1 / 0.6 / 1.1 / 1.0 / **0.8** | 0.0 / 0.9 / 1.7 / 1.6 / **1.7** | 0.1 / 0.9 / 2.5 / 2.7 / **2.9** |
| DI | 0.2 / 3.8 / 4.8 / 4.1 / **3.8** | 0.0 / 5.0 / 7.6 / 7.8 / **8.4** | 0.2 / 3.7 / 11.9 / 12.2 / **12.7** |
| SI | 1.3 / 9.0 / 8.9 / 7.7 / **7.2** | 0.3 / 14.0 / 15.6 / 16.4 / **16.8** | 1.3 / 8.2 / 17.0 / 17.4 / **17.8** |
| Admix | 4.9 / 14.3 / 13.4 / 13.2 / **13.0** | 0.7 / 17.9 / 20.5 / 20.2 / **20.2** | 4.9 / 14.0 / 23.9 / 24.2 / **23.6** |
| MI-TI-DI | 1.5 / 12.1 / 13.4 / 13.9 / **13.7** | 0.1 / 9.7 / 15.7 / 17.4 / **19.4** | 1.6 / 12.1 / 24.4 / 26.3 / **27.4** |
| MI-TI-DI-SI | 13.0 / 38.9 / 41.5 / 42.8 / **41.7** | 3.8 / 37.8 / 42.0 / 43.8 / **44.4** | 12.8 / 37.3 / 48.6 / 49.8 / **49.6** |
| MI-TI-DI-Admix | 19.0 / 45.5 / 47.0 / 47.7 / **48.0** | 6.8 / 41.3 / 45.2 / 44.8 / **45.1** | 19.1 / 45.3 / 52.9 / 54.9 / **55.2** |

| | Inc-v3 → ResNet-50 | | |
|---|---|---|---|
| | Baseline | +RAP | +RAP-LS |
| I | 0.2 / 0.4 / 0.3 / 0.1 / **0.2** | 0.0 / 0.2 / 0.7 / 0.6 / **0.9** | 0.2 / 0.4 / 1.0 / 0.7 / **0.5** |
| MI | 0.1 / 0.3 / 0.3 / 0.2 / **0.2** | 0.0 / 0.6 / 1.4 / 1.5 / **1.7** | 0.1 / 0.3 / 0.8 / 1.6 / **1.5** |
| TI | 0.2 / 0.3 / 0.2 / 0.2 / **0.2** | 0.0 / 0.2 / 0.6 / 0.9 / **0.5** | 0.2 / 0.3 / 1.0 / 0.7 / **0.7** |
| DI | 0.2 / 1.5 / 1.4 / 1.9 / **1.6** | 0.1 / 2.5 / 4.3 / 4.3 / **4.6** | 0.2 / 1.5 / 5.0 / 5.1 / **6.4** |
| SI | 0.3 / 0.3 / 0.3 / 0.6 / **0.6** | 0.4 / 1.9 / 2.6 / 2.6 / **2.9** | 0.3 / 0.3 / 2.4 / 2.8 / **2.5** |
| Admix | 1.2 / 1.9 / 2.2 / 1.9 / **1.5** | 0.6 / 5.0 / 4.9 / 5.2 / **4.9** | 1.2 / 1.9 / 5.7 / 5.7 / **5.2** |
| MI-TI-DI | 0.6 / 1.6 / 2.0 / 2.4 / **1.8** | 0.0 / 4.2 / 6.3 / 7.7 / **8.3** | 0.6 / 1.7 / 6.2 / 7.0 / **7.5** |
| MI-TI-DI-SI | 1.5 / 4.7 / 5.5 / 5.8 / **5.6** | 0.7 / 8.6 / 10.3 / 11.1 / **11.9** | 1.5 / 5.0 / 10.0 / 9.6 / **10.7** |
| MI-TI-DI-Admix | 2.8 / 8.9 / 9.5 / 9.6 / **9.6** | 1.4 / 12.6 / 14.0 / 13.6 / **13.6** | 2.8 / 8.6 / 14.5 / 15.1 / **16.7** |

| | Inc-v3 → DenseNet-121 | | |
|---|---|---|---|
| | Baseline | +RAP | +RAP-LS |
| I | 0.0 / 0.0 / 0.2 / 0.0 / **0.2** | 0.0 / 0.2 / 0.4 / 0.6 / **0.6** | 0.0 / 0.0 / 0.2 / 0.4 / **0.3** |
| MI | 0.0 / 0.1 / 0.2 / 0.1 / **0.1** | 0.1 / 0.7 / 1.0 / 1.1 / **1.6** | 0.0 / 0.1 / 1.0 / 1.1 / **1.5** |
| TI | 0.0 / 0.3 / 0.2 / 0.0 / **0.1** | 0.0 / 0.3 / 0.3 / 0.3 / **0.7** | 0.0 / 0.3 / 0.9 / 0.9 / **0.6** |
| DI | 0.1 / 1.3 / 2.5 / 3.0 / **2.8** | 0.0 / 2.7 / 4.4 / 5.4 / **5.8** | 0.1 / 1.3 / 5.9 / 7.0 / **7.5** |
| SI | 0.2 / 0.7 / 0.9 / 0.8 / **0.9** | 0.0 / 2.4 / 3.3 / 2.9 / **2.7** | 0.2 / 0.7 / 3.2 / 3.1 / **3.2** |
| Admix | 1.1 / 2.6 / 2.5 / 2.3 / **2.0** | 0.5 / 7.2 / 7.7 / 7.0 / **6.9** | 1.1 / 2.6 / 8.2 / 7.3 / **7.5** |
| MI-TI-DI | 0.5 / 3.1 / 3.8 / 4.5 / **4.1** | 0.2 / 5.4 / 10.8 / 12.6 / **14.8** | 0.5 / 3.3 / 10.6 / 11.8 / **13.4** |
| MI-TI-DI-SI | 1.9 / 9.0 / 9.4 / 9.5 / **10.4** | 1.1 / 15.5 / 19.8 / 19.8 / **21.2** | 1.9 / 9.0 / 19.1 / 20.2 / **20.9** |
| MI-TI-DI-Admix | 4.6 / 15.7 / 16.8 / 17.4 / **17.9** | 2.4 / 23.2 / 24.5 / 26.6 / **27.5** | 4.6 / 15.0 / 29.1 / 30.2 / **31.6** |

| | Inc-v3 → VGG-16 | | |
|---|---|---|---|
| | Baseline | +RAP | +RAP-LS |
| I | 0.0 / 0.3 / 0.1 / 0.1 / **0.1** | 0.0 / 0.2 / 0.8 / 0.6 / **0.5** | 0.0 / 0.3 / 0.2 / 0.5 / **0.5** |
| MI | 0.1 / 0.1 / 0.2 / 0.2 / **0.2** | 0.1 / 0.4 / 0.8 / 1.2 / **1.3** | 0.1 / 0.1 / 0.4 / 0.8 / **1.0** |
| TI | 0.1 / 0.2 / 0.2 / 0.1 / **0.2** | 0.1 / 0.4 / 0.5 / 0.6 / **0.8** | 0.1 / 0.2 / 0.6 / 0.6 / **0.6** |
| DI | 0.3 / 2.0 / 2.8 / 2.3 / **2.6** | 0.1 / 1.8 / 4.3 / 5.2 / **6.3** | 0.3 / 2.0 / 6.8 / 7.3 / **8.1** |
| SI | 0.0 / 0.7 / 0.6 / 0.4 / **0.5** | 0.2 / 2.0 / 1.5 / 1.5 / **1.5** | 0.0 / 0.7 / 1.6 / 2.3 / **2.3** |
| Admix | 0.5 / 1.6 / 1.0 / 1.0 / **1.3** | 0.4 / 3.2 / 3.8 / 4.1 / **3.3** | 0.5 / 1.6 / 4.5 / 3.8 / **4.4** |
| MI-TI-DI | 0.3 / 2.0 / 2.4 / 2.7 / **2.9** | 0.1 / 3.8 / 6.5 / 7.3 / **8.0** | 0.3 / 2.0 / 8.0 / 7.9 / **9.8** |
| MI-TI-DI-SI | 0.7 / 3.7 / 3.6 / 4.1 / **4.2** | 0.5 / 7.6 / 7.5 / 8.5 / **8.9** | 0.7 / 3.2 / 6.7 / 8.1 / **8.6** |
| MI-TI-DI-Admix | 2.3 / 6.9 / 8.3 / 8.6 / **8.4** | 1.3 / 10.5 / 12.5 / 12.0 / **12.0** | 2.3 / 7.0 / 11.9 / 12.8 / **12.1** |

Table 16: The untargeted attack success rate (%) of all baseline attacks with RAP. The results with $CE$ loss and 10/100/200/300/400 iterations are reported. We highlight the results with $K = 400$ in bold.

| | ResNet-50 → Inception-v3 | | |
| | Baseline | +RAP | +RAP-LS |
|---|---|---|---|
| I | 25.9 / 35.5 / 35.3 / 34.7 / **34.6** | 12.3 / 48.3 / 54.1 / 55.5 / **57.0** | 25.7 / 36.0 / 54.1 / 56.5 / **57.2** |
| MI | 53.2 / 50.7 / 51.0 / 50.6 / **50.3** | 26.2 / 58.7 / 68.9 / 73.4 / **75.9** | 53.2 / 50.7 / 64.3 / 73.6 / **77.4** |
| TI | 30.0 / 45.3 / 44.0 / 45.3 / **45.5** | 16.4 / 57.9 / 63.9 / 64.6 / **66.1** | 30.0 / 45.1 / 62.3 / 65.3 / **67.0** |
| DI | 46.0 / 60.5 / 59.5 / 59.4 / **57.7** | 27.3 / 80.7 / 82.8 / 83.4 / **82.9** | 46.0 / 61.0 / 86.0 / 85.7 / **85.0** |
| SI | 50.1 / 66.0 / 65.6 / 66.0 / **65.9** | 60.6 / 80.5 / 80.9 / 80.9 / **79.7** | 49.9 / 66.6 / 85.2 / 85.0 / **84.4** |
| Admix | 66.6 / 78.7 / 79.2 / 78.0 / **77.7** | 73.9 / 87.6 / 87.0 / 86.8 / **87.4** | 67.6 / 79.4 / 91.8 / 92.3 / **92.6** |
| MI-TI-DI | 82.1 / 85.8 / 86.4 / 85.9 / **85.7** | 61.9 / 93.9 / 95.3 / 95.6 / **96.0** | 82.1 / 85.8 / 95.9 / 96.4 / **96.9** |
| MI-TI-DI-SI | 94.2 / 96.8 / 97.2 / 97.0 / **97.0** | 92.3 / 98.9 / 98.9 / 99.0 / **99.1** | 94.2 / 96.7 / 99.0 / 99.3 / **99.1** |
| MI-TI-DI-Admix | 97.3 / 98.6 / 98.5 / 98.5 / **98.3** | 95.1 / 99.4 / 99.4 / 99.3 / **99.2** | 97.3 / 98.5 / 99.8 / 99.8 / **99.8** |

| | ResNet-50 → DenseNet-121 | | |
| | Baseline | +RAP | +RAP-LS |
|---|---|---|---|
| I | 67.4 / 79.9 / 79.1 / 79.0 / **79.2** | 26.7 / 84.8 / 91.1 / 90.8 / **91.5** | 67.8 / 80.1 / 89.8 / 91.3 / **91.9** |
| MI | 87.3 / 85.4 / 86.4 / 85.9 / **85.8** | 45.2 / 85.3 / 91.3 / 93.9 / **95.0** | 87.3 / 85.4 / 90.8 / 95.0 / **96.1** |
| TI | 73.2 / 83.0 / 82.2 / 81.6 / **82.0** | 30.9 / 87.3 / 91.5 / 93.3 / **94.1** | 72.9 / 82.4 / 90.9 / 94.2 / **95.1** |
| DI | 92.8 / 98.9 / 99.2 / 99.0 / **99.0** | 52.6 / 99.0 / 99.6 / 99.7 / **99.6** | 92.8 / 99.0 / 99.6 / 99.7 / **99.7** |
| SI | 89.1 / 95.7 / 95.6 / 95.3 / **94.9** | 91.3 / 98.9 / 99.0 / 99.2 / **98.9** | 89.1 / 95.7 / 99.7 / 99.7 / **99.7** |
| Admix | 96.6 / 98.9 / 98.5 / 98.1 / **97.9** | 96.2 / 99.6 / 99.6 / 99.6 / **99.6** | 96.4 / 98.5 / 99.9 / 99.9 / **99.9** |
| MI-TI-DI | 98.2 / 99.7 / 99.8 / 99.8 / **99.8** | 86.4 / 99.9 / 100 / 100 / **100** | 98.2 / 99.7 / 99.9 / 100 / **100** |
| MI-TI-DI-SI | 99.8 / 100 / 100 / 100 / **100** | 98.8 / 100 / 100 / 100 / **100** | 99.8 / 100 / 100 / 100 / **100** |
| MI-TI-DI-Admix | 99.9 / 100 / 100 / 100 / **100** | 99.5 / 100 / 100 / 100 / **100** | 99.9 / 100 / 100 / 100 / **100** |

| | ResNet-50 → VGG-16 | | |
| | Baseline | +RAP | +RAP-LS |
|---|---|---|---|
| I | 68.2 / 77.4 / 78.1 / 77.4 / **78.0** | 36.2 / 84.6 / 89.2 / 90.7 / **91.1** | 68.4 / 77.3 / 87.1 / 90.9 / **92.9** |
| MI | 82.5 / 82.8 / 82.9 / 82.7 / **82.4** | 53.1 / 85.5 / 92.2 / 93.1 / **93.9** | 82.5 / 82.8 / 89.3 / 93.7 / **94.5** |
| TI | 70.6 / 80.5 / 79.8 / 80.8 / **81.0** | 39.3 / 86.9 / 90.6 / 92.5 / **93.1** | 71.1 / 80.0 / 89.0 / 91.9 / **93.3** |
| DI | 92.3 / 99.1 / 99.1 / 99.0 / **99.0** | 64.4 / 99.4 / 99.7 / 99.7 / **99.6** | 92.3 / 99.1 / 99.8 / 99.9 / **99.7** |
| SI | 82.2 / 90.0 / 88.9 / 89.6 / **88.6** | 81.3 / 95.7 / 95.8 / 95.7 / **95.7** | 82.1 / 89.3 / 97.7 / 97.8 / **97.2** |
| Admix | 92.3 / 95.4 / 96.0 / 95.6 / **95.8** | 91.6 / 97.9 / 98.4 / 97.8 / **97.7** | 92.7 / 95.9 / 98.9 / 99.0 / **99.0** |
| MI-TI-DI | 97.9 / 99.7 / 99.7 / 99.8 / **99.8** | 85.9 / 99.5 / 100 / 100 / **100** | 97.9 / 99.7 / 99.9 / 99.9 / **99.9** |
| MI-TI-DI-SI | 99.1 / 99.8 / 99.8 / 99.7 / **99.7** | 97.4 / 99.7 / 99.9 / 99.9 / **99.9** | 99.1 / 99.8 / 99.8 / 99.8 / **99.8** |
| MI-TI-DI-Admix | 99.2 / 99.8 / 99.8 / 99.8 / **99.8** | 98.5 / 99.7 / 99.9 / 99.9 / **99.9** | 99.2 / 99.8 / 99.9 / 99.9 / **99.9** |

| | DenseNet-121 → Inception-v3 | | |
| | Baseline | +RAP | +RAP-LS |
|---|---|---|---|
| I | 31.2 / 48.5 / 46.9 / 46.3 / **46.5** | 18.0 / 54.9 / 58.1 / 59.8 / **60.2** | 31.6 / 46.9 / 58.9 / 61.0 / **61.1** |
| MI | 56.8 / 58.8 / 59.3 / 60.6 / **59.3** | 32.2 / 65.6 / 74.1 / 78.9 / **80.4** | 56.8 / 58.8 / 74.6 / 80.0 / **82.8** |
| TI | 37.7 / 54.0 / 55.1 / 54.6 / **54.2** | 20.4 / 61.0 / 64.7 / 67.3 / **66.7** | 38.2 / 54.5 / 65.4 / 67.6 / **70.0** |
| DI | 51.0 / 67.9 / 68.3 / 66.7 / **67.6** | 31.4 / 84.0 / 86.8 / 86.7 / **86.6** | 51.0 / 68.0 / 89.0 / 88.8 / **86.9** |
| SI | 54.7 / 71.5 / 71.6 / 70.3 / **71.6** | 61.1 / 82.9 / 83.1 / 83.5 / **83.2** | 53.9 / 71.0 / 86.4 / 87.0 / **87.4** |
| Admix | 72.5 / 82.0 / 82.6 / 82.2 / **82.0** | 73.0 / 89.9 / 90.3 / 89.5 / **89.8** | 71.7 / 82.8 / 93.9 / 93.2 / **93.8** |
| MI-TI-DI | 81.5 / 89.7 / 89.8 / 89.4 / **89.1** | 62.5 / 94.8 / 96.8 / 97.1 / **97.1** | 81.5 / 89.6 / 96.1 / 96.9 / **97.1** |
| MI-TI-DI-SI | 92.3 / 95.2 / 94.9 / 95.1 / **95.1** | 88.6 / 97.7 / 98.0 / 98.0 / **98.3** | 92.4 / 95.2 / 97.8 / 98.5 / **98.4** |
| MI-TI-DI-Admix | 95.8 / 97.7 / 97.2 / 97.3 / **97.9** | 93.2 / 98.6 / 98.6 / 99.0 / **98.8** | 95.4 / 97.6 / 99.0 / 98.9 / **98.9** |

| | DenseNet-121 → ResNet-50 | | |
| | Baseline | +RAP | +RAP-LS |
|---|---|---|---|
| I | 76.1 / 88.0 / 87.5 / 87.1 / **87.4** | 35.7 / 90.1 / 93.5 / 93.2 / **94.2** | 76.1 / 88.0 / 91.2 / 92.9 / **94.3** |
| MI | 87.7 / 90.5 / 91.2 / 90.8 / **90.3** | 55.6 / 91.1 / 96.2 / 96.9 / **97.6** | 87.7 / 90.5 / 95.4 / 97.2 / **97.9** |
| TI | 79.2 / 90.4 / 90.0 / 89.9 / **89.6** | 36.9 / 90.1 / 93.2 / 95.0 / **94.2** | 79.0 / 89.8 / 92.7 / 94.3 / **94.8** |
| DI | 91.1 / 98.0 / 98.3 / 98.2 / **98.2** | 57.0 / 98.6 / 99.3 / 99.7 / **99.6** | 91.1 / 98.0 / 99.5 / 99.6 / **99.7** |
| SI | 89.6 / 95.2 / 94.8 / 95.3 / **95.1** | 83.0 / 96.5 / 96.7 / 96.3 / **96.9** | 89.4 / 95.0 / 98.7 / 98.8 / **98.8** |
| Admix | 96.3 / 97.6 / 97.7 / 97.7 / **97.0** | 90.9 / 98.8 / 98.8 / 99.0 / **99.0** | 95.7 / 97.9 / 99.3 / 99.2 / **99.2** |
| MI-TI-DI | 96.3 / 99.3 / 99.5 / 99.4 / **99.4** | 84.4 / 99.2 / 99.8 / 99.8 / **99.8** | 96.3 / 99.2 / 99.8 / 99.9 / **100** |
| MI-TI-DI-SI | 98.3 / 99.7 / 99.8 / 99.8 / **99.8** | 95.8 / 99.7 / 99.9 / 99.9 / **99.9** | 98.3 / 99.7 / 99.9 / 99.9 / **99.9** |
| MI-TI-DI-Admix | 99.2 / 99.7 / 99.8 / 99.8 / **99.8** | 97.9 / 99.9 / 99.8 / 99.8 / **99.8** | 99.0 / 99.7 / 99.9 / 99.9 / **99.9** |

| | DenseNet-121 → VGG-16 | | |
| --- | --- | --- | --- |
| | Baseline | +RAP | +RAP-LS |
| I | 75.1 / 84.7 / 85.2 / 84.9 / **85.1** | 42.2 / 87.5 / 90.7 / 91.2 / **91.7** | 75.1 / 84.6 / 89.2 / 91.7 / **92.8** |
| MI | 85.1 / 87.2 / 88.6 / 87.9 / **87.5** | 58.4 / 90.2 / 93.7 / 95.1 / **96.0** | 85.1 / 87.2 / 94.2 / 97.0 / **97.6** |
| TI | 74.4 / 86.3 / 86.4 / 87.3 / **87.0** | 44.2 / 87.8 / 89.6 / 91.0 / **92.1** | 74.5 / 85.8 / 90.3 / 92.2 / **93.3** |
| DI | 90.8 / 98.0 / 98.4 / 98.1 / **98.1** | 63.3 / 98.6 / 99.2 / 99.6 / **99.4** | 90.8 / 97.9 / 99.4 / 99.2 / **99.4** |
| SI | 84.2 / 91.5 / 91.4 / 91.4 / **91.9** | 78.5 / 93.9 / 94.5 / 95.2 / **95.0** | 83.9 / 91.6 / 96.9 / 97.1 / **97.5** |
| Admix | 93.5 / 95.7 / 96.0 / 96.1 / **95.6** | 87.8 / 97.4 / 97.5 / 97.6 / **97.7** | 92.0 / 96.1 / 98.9 / 98.7 / **98.6** |
| MI-TI-DI | 95.1 / 99.0 / 99.2 / 99.2 / **99.2** | 84.2 / 99.1 / 99.4 / 99.5 / **99.5** | 95.1 / 99.0 / 99.9 / 100 / **100** |
| MI-TI-DI-SI | 97.9 / 99.5 / 99.4 / 99.4 / **99.2** | 93.3 / 99.0 / 99.2 / 99.3 / **99.3** | 97.9 / 99.4 / 99.7 / 99.7 / **99.7** |
| MI-TI-DI-Admix | 98.4 / 99.4 / 99.4 / 99.5 / **99.4** | 96.1 / 99.7 / 99.7 / 99.6 / **99.6** | 98.3 / 99.4 / 99.8 / 99.7 / **99.8** |

| | VGG-16 → Inception-v3 | | |
| --- | --- | --- | --- |
| | Baseline | +RAP | +RAP-LS |
| I | 14.3 / 22.2 / 22.0 / 22.2 / **22.0** | 9.4 / 23.8 / 26.1 / 23.7 / **24.7** | 14.4 / 21.8 / 24.1 / 25.4 / **24.9** |
| MI | 32.3 / 31.3 / 31.0 / 30.1 / **30.0** | 16.4 / 30.4 / 36.9 / 42.0 / **42.7** | 32.4 / 30.7 / 35.0 / 39.2 / **42.2** |
| TI | 18.7 / 30.2 / 29.6 / 29.7 / **29.1** | 11.9 / 32.1 / 35.7 / 34.9 / **36.2** | 18.3 / 29.3 / 34.2 / 36.0 / **37.1** |
| DI | 18.1 / 29.7 / 29.9 / 30.4 / **29.9** | 14.2 / 43.6 / 46.1 / 46.5 / **46.6** | 18.0 / 29.2 / 50.1 / 51.5 / **51.6** |
| SI | 31.0 / 45.1 / 46.1 / 45.1 / **45.8** | 46.7 / 70.9 / 72.0 / 73.4 / **74.0** | 31.0 / 44.6 / 73.0 / 74.3 / **74.7** |
| Admix | 40.2 / 54.9 / 55.5 / 54.9 / **55.5** | 57.0 / 78.0 / 77.6 / 77.9 / **77.6** | 41.4 / 56.0 / 80.0 / 79.9 / **80.8** |
| MI-TI-DI | 50.7 / 55.9 / 57.2 / 56.7 / **56.8** | 41.9 / 74.0 / 79.0 / 81.5 / **82.6** | 50.7 / 56.4 / 77.8 / 80.0 / **81.4** |
| MI-TI-DI-SI | 77.6 / 85.3 / 85.7 / 85.0 / **85.0** | 85.5 / 93.1 / 93.7 / 94.2 / **94.1** | 78.0 / 85.0 / 94.4 / 94.6 / **95.2** |
| MI-TI-DI-Admix | 84.7 / 89.4 / 89.2 / 89.9 / **89.3** | 88.4 / 94.9 / 95.1 / 95.2 / **95.0** | 85.8 / 90.1 / 94.8 / 95.4 / **95.5** |

| | VGG-16 → ResNet-50 | | |
| --- | --- | --- | --- |
| | Baseline | +RAP | +RAP-LS |
| I | 37.2 / 52.0 / 53.4 / 53.1 / **53.7** | 17.8 / 48.5 / 53.9 / 53.7 / **53.0** | 38.1 / 53.0 / 52.4 / 54.8 / **54.2** |
| MI | 60.2 / 64.3 / 63.5 / 62.0 / **62.5** | 32.9 / 57.1 / 67.6 / 73.1 / **76.2** | 60.4 / 62.0 / 66.3 / 73.2 / **76.4** |
| TI | 45.3 / 62.7 / 63.6 / 62.5 / **62.8** | 19.4 / 56.6 / 63.0 / 65.6 / **64.8** | 46.0 / 62.9 / 63.5 / 65.8 / **65.8** |
| DI | 51.5 / 72.9 / 73.2 / 72.5 / **72.2** | 29.6 / 80.9 / 85.0 / 86.4 / **86.0** | 51.4 / 73.8 / 88.9 / 89.2 / **88.8** |
| SI | 64.6 / 81.0 / 80.2 / 80.5 / **80.0** | 68.1 / 91.9 / 92.3 / 92.4 / **92.7** | 64.9 / 80.6 / 95.1 / 95.3 / **94.7** |
| Admix | 76.8 / 87.5 / 88.2 / 88.0 / **87.3** | 79.4 / 93.8 / 94.4 / 95.2 / **94.6** | 77.6 / 88.3 / 96.6 / 96.8 / **96.8** |
| MI-TI-DI | 81.1 / 89.9 / 89.8 / 90.3 / **90.0** | 66.7 / 94.6 / 96.3 / 96.9 / **97.2** | 81.4 / 88.5 / 96.5 / 97.3 / **97.7** |
| MI-TI-DI-SI | 95.1 / 97.6 / 98.0 / 97.9 / **97.6** | 94.7 / 98.4 / 98.8 / 98.9 / **98.8** | 95.2 / 97.5 / 99.3 / 99.4 / **99.4** |
| MI-TI-DI-Admix | 97.2 / 98.1 / 98.0 / 98.1 / **97.8** | 96.1 / 99.1 / 99.2 / 99.3 / **99.2** | 97.3 / 98.6 / 99.5 / 99.6 / **99.6** |

| | VGG-16 → DenseNet-121 | | |
| --- | --- | --- | --- |
| | Baseline | +RAP | +RAP-LS |
| I | 35.4 / 50.4 / 49.8 / 48.4 / **49.1** | 15.4 / 46.0 / 49.6 / 50.5 / **50.6** | 35.2 / 50.3 / 49.7 / 52.9 / **51.4** |
| MI | 62.1 / 63.8 / 62.8 / 61.7 / **60.5** | 26.6 / 51.1 / 63.4 / 70.0 / **73.0** | 61.6 / 62.5 / 62.7 / 70.5 / **73.9** |
| TI | 43.5 / 58.6 / 58.7 / 57.2 / **55.9** | 19.4 / 55.8 / 62.7 / 63.0 / **63.7** | 44.3 / 58.3 / 60.3 / 63.8 / **62.1** |
| DI | 48.1 / 70.2 / 68.9 / 70.0 / **68.8** | 26.5 / 79.9 / 82.3 / 84.2 / **85.0** | 47.9 / 70.5 / 85.1 / 87.2 / **87.2** |
| SI | 65.3 / 82.3 / 82.4 / 82.0 / **82.1** | 71.3 / 93.3 / 93.7 / 94.4 / **94.8** | 65.5 / 82.2 / 95.2 / 95.4 / **95.7** |
| Admix | 79.6 / 89.4 / 88.6 / 88.4 / **88.2** | 83.5 / 96.1 / 95.9 / 96.2 / **96.4** | 79.2 / 88.9 / 97.4 / 97.4 / **97.2** |
| MI-TI-DI | 80.3 / 87.0 / 88.7 / 89.3 / **88.8** | 62.9 / 94.0 / 95.9 / 96.4 / **97.0** | 80.4 / 86.8 / 96.8 / 97.2 / **97.3** |
| MI-TI-DI-SI | 95.3 / 98.2 / 98.4 / 98.4 / **98.1** | 95.9 / 99.2 / 99.2 / 99.2 / **99.2** | 95.4 / 98.2 / 99.5 / 99.5 / **99.4** |
| MI-TI-DI-Admix | 97.1 / 98.6 / 98.8 / 99.1 / **98.9** | 97.4 / 99.4 / 99.6 / 99.6 / **99.5** | 97.3 / 98.5 / 99.5 / 99.5 / **99.6** |

| | Inc-v3 → ResNet-50 | | |
| --- | --- | --- | --- |
| | Baseline | +RAP | +RAP-LS |
| I | 34.0 / 48.4 / 51.2 / 50.1 / **51.5** | 22.7 / 58.6 / 60.9 / 61.1 / **62.1** | 34.5 / 49.0 / 60.2 / 60.5 / **62.0** |
| MI | 58.5 / 59.1 / 60.4 / 60.3 / **62.0** | 43.8 / 77.0 / 81.7 / 84.0 / **85.8** | 58.5 / 59.1 / 80.0 / 82.6 / **84.8** |
| TI | 33.6 / 46.9 / 48.7 / 48.5 / **49.3** | 21.8 / 58.9 / 60.2 / 61.7 / **63.4** | 33.1 / 47.2 / 59.5 / 61.5 / **61.6** |
| DI | 48.4 / 65.8 / 67.2 / 68.4 / **68.4** | 33.3 / 78.8 / 81.4 / 81.4 / **81.7** | 48.3 / 65.7 / 80.7 / 82.3 / **81.8** |
| SI | 43.7 / 61.9 / 63.9 / 65.1 / **66.2** | 45.8 / 67.0 / 69.4 / 69.5 / **69.8** | 43.6 / 62.3 / 72.5 / 73.4 / **72.8** |
| Admix | 56.1 / 73.0 / 75.9 / 76.9 / **75.9** | 57.0 / 77.5 / 79.8 / 80.3 / **80.2** | 56.3 / 73.4 / 82.9 / 84.0 / **84.9** |
| MI-TI-DI | 72.2 / 79.5 / 81.9 / 81.9 / **82.9** | 61.2 / 88.1 / 90.8 / 91.9 / **91.8** | 72.2 / 79.4 / 89.9 / 91.5 / **90.6** |
| MI-TI-DI-SI | 82.9 / 88.3 / 88.3 / 88.4 / **89.0** | 83.5 / 90.8 / 91.2 / 90.6 / **91.2** | 82.8 / 88.1 / 91.9 / 92.6 / **92.3** |
| MI-TI-DI-Admix | 89.8 / 91.6 / 91.3 / 91.4 / **91.5** | 89.0 / 93.9 / 94.0 / 94.0 / **94.1** | 89.6 / 92.3 / 94.1 / 94.8 / **94.7** |

| | Inc-v3 → DenseNet-121 | | |
| | Baseline | +RAP | +RAP-LS |
|---|---|---|---|
| I | 35.2 / 47.2 / 47.3 / 46.8 / **48.7** | 21.4 / 54.3 / 57.2 / 59.4 / **60.8** | 34.9 / 47.5 / 58.7 / 58.8 / **60.0** |
| MI | 57.4 / 56.2 / 56.5 / 56.8 / **56.7** | 42.9 / 74.0 / 80.1 / 82.5 / **84.6** | 57.4 / 56.2 / 77.4 / 81.9 / **84.6** |
| TI | 35.8 / 48.6 / 47.8 / 48.9 / **49.4** | 22.1 / 59.6 / 63.3 / 65.7 / **63.4** | 35.5 / 48.7 / 61.6 / 64.2 / **63.8** |
| DI | 53.2 / 72.1 / 71.8 / 71.5 / **71.9** | 35.7 / 81.9 / 83.7 / 85.1 / **85.0** | 53.2 / 71.8 / 84.1 / 85.2 / **84.0** |
| SI | 46.6 / 63.7 / 65.1 / 65.9 / **65.9** | 52.6 / 72.4 / 73.5 / 74.5 / **74.9** | 46.6 / 63.0 / 77.7 / 77.9 / **77.2** |
| Admix | 60.5 / 76.7 / 78.0 / 79.3 / **78.5** | 63.9 / 83.2 / 83.4 / 84.1 / **83.7** | 61.9 / 76.9 / 87.7 / 87.3 / **87.4** |
| MI-TI-DI | 76.7 / 84.7 / 85.7 / 85.7 / **85.7** | 65.1 / 91.5 / 92.8 / 94.0 / **94.2** | 76.7 / 84.6 / 92.6 / 92.9 / **93.3** |
| MI-TI-DI-SI | 89.0 / 91.9 / 91.7 / 91.8 / **92.0** | 89.0 / 94.7 / 95.6 / 95.2 / **95.2** | 89.0 / 91.4 / 95.1 / 95.4 / **95.6** |
| MI-TI-DI-Admix | 93.5 / 95.5 / 95.9 / 95.1 / **95.4** | 93.3 / 96.8 / 96.9 / 96.4 / **96.2** | 94.1 / 95.5 / 97.2 / 97.5 / **97.6** |

| | Inc-v3 → VGG-16 | | |
| | Baseline | +RAP | +RAP-LS |
|---|---|---|---|
| I | 39.9 / 53.1 / 54.1 / 53.7 / **55.1** | 29.1 / 63.0 / 65.8 / 66.9 / **65.9** | 39.7 / 52.6 / 65.6 / 68.3 / **68.0** |
| MI | 60.7 / 62.2 / 63.8 / 62.1 / **63.1** | 50.7 / 76.1 / 81.0 / 83.6 / **84.9** | 60.7 / 62.2 / 79.8 / 84.0 / **84.6** |
| TI | 41.6 / 55.1 / 55.2 / 55.3 / **58.1** | 31.1 / 65.9 / 67.1 / 68.2 / **68.6** | 41.5 / 55.1 / 66.3 / 68.0 / **69.5** |
| DI | 54.9 / 73.4 / 74.5 / 76.0 / **76.1** | 44.4 / 83.4 / 84.7 / 85.0 / **85.2** | 54.9 / 73.0 / 85.7 / 87.2 / **86.4** |
| SI | 46.7 / 62.4 / 64.4 / 65.7 / **66.0** | 47.4 / 67.6 / 69.2 / 68.6 / **69.2** | 46.3 / 64.1 / 72.4 / 72.1 / **73.0** |
| Admix | 57.3 / 73.2 / 72.8 / 74.0 / **74.5** | 57.3 / 75.4 / 75.9 / 77.5 / **77.2** | 55.3 / 73.4 / 82.6 / 82.2 / **83.5** |
| MI-TI-DI | 74.7 / 82.7 / 84.7 / 84.6 / **85.1** | 67.7 / 90.0 / 91.9 / 92.3 / **92.7** | 74.7 / 82.5 / 90.4 / 90.8 / **91.0** |
| MI-TI-DI-SI | 79.8 / 88.0 / 87.6 / 87.5 / **87.6** | 81.6 / 89.0 / 89.4 / 89.4 / **90.3** | 79.7 / 87.8 / 92.4 / 92.5 / **92.9** |
| MI-TI-DI-Admix | 87.9 / 89.7 / 90.7 / 91.4 / **91.4** | 87.0 / 92.2 / 92.3 / 92.5 / **93.2** | 87.7 / 91.7 / 94.5 / 94.6 / **94.1** |

