# OpenReview forum: "Boosting the Transferability of Adversarial Attacks with Reverse Adversarial Perturbation"
_ICLR.cc/2022/Conference — ICLR 2022 Submitted_

### Official Review · Reviewer_PVqn · 2021-10-31

**Correctness:** 3
**Technical Novelty And Significance:** 2
**Empirical Novelty And Significance:** 4
**Recommendation:** 6
**Confidence:** 3

**Main Review:**

Strengths:

+ The bi-level min-max formulation is novel for generating transferrable attacks.
+ There exist extensive experiments involving a variety of model architectures and baselines.


Weakness:

- The flatness of loss landscape with respect to (w.r.t.) input vs. model generalization should be more carefully studied. My doubt arises from the paper [https://openreview.net/pdf?id=BylKL1SKvr]. This work showed that model transferability (from the source domain and the target domain) relates to the smoothness of loss landscape, but this conclusion holds for loss landscape w.r.t. model parameters. I understood why authors draw a connection between the flatness of loss landscape and model generalization in the context of attack generation. However, has this been well studied and believed as a grounded conclusion?

- The use of a min-max formulation for attack generation has been studied in [https://arxiv.org/pdf/1906.03563.pdf]. This related work should be covered and discussed in Related Work.

- If the min-max formulation (2) is replaced with EoT (expectation over transformation), namely, n^{adv} is randomly generated from e.g., Gaussian distribution, then this will lead to an EoT-type baseline. It will be better to cover this baseline as well to demonstrate the superiority of bi-level formulation.

- Min-max attack generation could be difficult to tune. Thus, the computation time and hyper-parameter setups of the proposed approach should be clearly stated.

- If the n^{adv} is regarded as model noise to flatten the loss landscape, then how does the attack perform compared to RAP? This is another baseline to verify the usefulness of the flatness of loss landscape w.r.t. `input' rather than 'model parameters'.







**Summary Of The Paper:**

This paper proposed a min-max formulation to improve attack transferability. The key idea is motivated by the fact that the smoothness of the loss landscape could improve model generalization ability.  Thus, a reverse adversarial noise (for landscape smoothing) is injected so as to compensate for the effect of an adversarial attack. Numerous experiments are provided to demonstrate the effectiveness of reverse adversarial perturbation (RAP).



**Summary Of The Review:**

I think this is an Okay submission, but several technical and experimental questions remain.

---

> ### Author Response · Authors · 2021-11-22
> **Response to Reviewer PVqn (Part 1/2)**
>
> ### **Q4.1**: "The flatness of loss landscape with respect to (w.r.t.) input vs. model ...... generalization in the context of attack generation. However, has this been well studied and believed as a grounded conclusion?"
>
> ### **R4.1**:
>
> Thanks for this constructive comment.
> The adversarial transferability is different from model generalization or model transferability. The former studies the transferability of the given adversarial example $x^{adv}$ across the different models. While the latter studies the transferability of the given model across the various datasets.
> Our work is inspired by the connections between model generalization and flatter local minimal in standard training, which has been studied very well in existing works, including https://openreview.net/pdf?id=BylKL1SKvr.
>
> The positive connection between flatter loss landscape w.r.t. $x^{adv}$ and the attack transferability is not a grounded conclusion yet. Actually, analyzing the flatness of loss landscape w.r.t. $x^{adv}$ and attack transferability is one of our contributions and we found few works considering this. We have verified this experimentally in the current submission and will explore more in the future.
>
> ----
>
> ### **Q4.2**: "The use of a min-max formulation for attack generation has been studied in [https://arxiv.org/pdf/1906.03563.pdf]. This related work should be covered and discussed in Related Work."
>
>
> ### **R4.2**:
>
> Thanks for pointing out this. Actually, the work of https://arxiv.org/pdf/1906.03563.pdf [1] has been put on Arxiv since two years ago and is recently updated to NeurIPS 2021 version after we submitted our work. Although the work of [1] also explores the min-max framework for producing the attacks, our method is totally different from it in terms of motivations and formulations.
>
> Our work is based on the assumption that the flat landscape could lead to better generalization. Therefore, we proposed to minimize the maximal loss value within a local neighborhood region around the adversarial example $x^{adv}$. The variable of our inner maximization is sample perturbation. The work of [1] aims to generate attacks based on multiple models or multiple samples. And, the variable of inner maximization is a probability vector $w$, representing the weight assigned to different models or samples.
>
> We have added these discussions into Section 2 Related work in the revised version.
>
> [1] Adversarial Attack Generation Empowered by Min-Max Optimization, NeurIPS 2021
>
> ----
>
> ### **Q4.3**: "If the min-max formulation (2) is replaced with EoT (expectation over transformation), namely, n^{adv} is randomly generated from e.g., Gaussian distribution, then this will lead to an EoT-type baseline. It will be better to cover this baseline as well to demonstrate the superiority of bi-level formulation."
>
> ### **R4.3**:
>
> Thanks for this useful suggestion. We conducted experiments of the suggested baseline. In each iteration, we sample random noise $n^{adv}$ from Gaussian distribution multiple times and add them to $x^{adv}$ following the expectation of transformation (EOT).  We set the standard deviation of Gaussian noise as $\sigma = 15/255$ and the number of sampling as $k=20$, which performs best in our ablation study of $\sigma = (1/255，5/255，10/255，15/255，25/255)$ and $k=(5,10,20,30)$. We take MTDI as our baseline, ResNet and DenseNet as surrogate models and compare this attack and our method RAP.
>
> **Experimental Results**: The untargeted and targeted attack performance is shown in the below table. As shown in experimental results, our RAP-LS achieves better performance and surpasses the EOT attack by a large margin for targeted attacks. RAP-LS gets the increase by 8.7% for targeted attacks in terms of average success rate. This demonstrates that the superiority of our methods.
>
> *Table 1: The untargted and targeted attack success rates (%). The baseline method is MI-TI-DI (MTDI).*
>
> |       Attack ($\mathcal{M}^{s}$: Res-50)      | \| Inc-v3 | \| Dense-121 |  \| VGG-16   |
> | :----------------: | :----------: | :----------: | :---------: |
> | MTDI+EOT / MTDI + RAP-LS (Untargeted) | \| 95.3 / 96.9  |  \| 99.9 / 100  | \| 99.7 / 99.9 |
> |  MTDI+EOT / MTDI + RAP-LS (Targeted)  | \| 34.6 / 33.2  | \| 82.7 / 88.5  | \| 73.4 / 81.5 |
>
>
> |       Attack ($\mathcal{M}^{s}$: Dense-121)      | \| Inc-v3 |  \| Res-50  |  \| VGG-16   |
> | :----------------: | :----------: | :---------: | :---------: |
> | MTDI+EOT / MTDI + RAP-LS (Untargeted) | \| 96.3 / 97.1  | \| 99.8 / 100  | \| 99.7 / 100  |
> |  MTDI+EOT / MTDI + RAP-LS (Targeted)  | \| 22.6 / 26.5  | \| 56.6 / 74.5 | \| 49.7 / 65.5 |

---

> > ### Comment · Reviewer_PVqn · 2021-11-24
> > **Thanks for the response**
> >
> > Most of my previous concerns have been properly addressed, except the connection between the flatness of loss landscape with respect to (w.r.t.) input and model generalization.
> >
> > In Sec. 3.2, it stated that "Recalling that prior works have related the model generalization to the flatness (or sharpness) of the loss landscape in the weight space, motivating a series of methods on boosting the model generalization by implicitly or explicitly seeking a flatter minima" and "Inspired by this point, we assume that the adversarial example locating at flatter regions of the loss landscape of the surrogate model may have better adversarial transferability to other models."
> >
> > However, the above connection 'model generalization-flatness of loss landscape-attack transferability' is not tightly coupled. The reason is that model generalization relates to the flatness of loss landscape w.r.t. model parameters rather than input variables. This causes confusion.
> >
> > If the connection is sound, then does this mean that generating adversarial examples from a more transferable surrogate model (across datasets) will lead to better attack transferability to the other victim models?
> >
> > I will consider increasing my score if a clearer connection can be drawn.

---

> > > ### Author Response · Authors · 2021-11-25
> > > **The new response to Reviewer PVqn (Part 1/2)**
> > >
> > > We are encouraged by your positive comments on most parts of our responses. For the remaining concern, we try to clarify it from the following two aspects.
> > >
> > > 1) **Different perspectives of the flatness of loss landscape *w.r.t.* input and weight.**
> > >     Given the training dataset $D=${$x_{i},y_{i}$}, the loss function $L$ and the classification model $\mathcal{M}$ parameterized by $\theta$, the loss landscape is denoted as $\mathcal{L}(D, \theta) = \frac{1}{n} \sum_{i=1}^{n} L(\mathcal{M}(x_{i};\theta),y_i)$. There are two different perspectives of $\mathcal{L}(D, \theta)$ to understand the behind meaning of its flatness.
> > >
> > >     - **Perspective I: The loss landscape *w.r.t.* weight $\theta$** describes that given a fixed training dataset $D$,  how $\mathcal{L}(D, \theta)$ varies *w.r.t.* $\theta$. In the standard supervised learning, one aims to find a suitable $\theta$ to fit the data $D$ well, by minimizing $\mathcal{L}(D, \theta)$. However, if $D$ is fitted very well, the loss landscape may become very fluctuating (*i.e.*, not flat) *w.r.t.* $\theta$, also called **overfitting**. Consequently, the learned model $\mathcal{M}_{\theta}$ may fit unseen data very poor, *i.e.*, poor generalization to new dataset. Thus, the flatness of $\mathcal{L}(D, \theta)$ w.r.t. $\theta$ is usually desired to mitigate the overfitting to $D$. The global flatness can be simply achieved by adding a regularization term (*e.g.*, $|| \theta ||^2$), while the flatness in local region can be implemented by slightly perturbing $\theta$ during the learning via an inner maximization, as did in a recent work [1].
> > >     - **Perspective II: The loss landscape w.r.t input $(x,y)$** describes that given a fixed model $\mathcal{M}_{\theta}$, how $\mathcal{L}((x,y), \theta)$ varies *w.r.t.* $x$. The goal is to find a suitable $x$ to fit the weight $\theta$ very well, by minimizing $\mathcal{L}((x,y), \theta)$. However, analogue to Persective I, if $\theta$ is fitted very well, then the obtained $x$ may fit unseen new model weight very poor. Thus, the flatness of $\mathcal{L}((x,y), \theta)$ *w.r.t.*  $x$ is also desired to mitigate the overfitting to $\theta$. In other words, a desired $x$ should locate at a flat and low-loss region.
> > >     In the scenario of adversarial transferability, our goal is to find a suitable $x^{adv}$ in the vicinity of the benign input $x$ that can not only fit the surrogate model $\theta^s$ (*i.e.*, minimizing the loss $\mathcal{L}((x^{adv},y_t), \theta^s)$), but also fit the unseen target model $\theta^t$ well (*i.e.*, good adversarial transferability). Such a $x^{adv}$ is expected to locate at a flat region in the landscape $\mathcal{L}((x^{adv},y_t), \theta^s)$. This is the main motivation of our min-max formulation, which can achieve this goal, as visualized in Figure 2 of our original manuscript and empirically verified by our extentive experiments.
> > >     Besides, another intuitive interpretation behind our method is that by pursuing $x$ locating at a flat local region, one could obtain a more robust adversarial example, *i.e.*, not only $x^{adv}$ is adversarial, but also the neighbors around $x^{adv}$. It could provide some tolerance for weight fluctuations between source and target models so as to boost transferability.

---

> > > ### Author Response · Authors · 2021-11-25
> > > **The new response to Reviewer PVqn (Part 2/2)**
> > >
> > > 2) **''Will generating adversarial examples from a more transferable surrogate model (across datasets) lead to better attack transferability to the other victim models?''**
> > >      This is a very insightful idea.
> > >      -  We also agree that designing more transferable surrogate models is a promising approach to boost adversarial transferability. The key point is how to measure the transferability of a model. A recent work [2] attempted to measure the model transferability based on the connection between adversarial transferability and knowledge transferability (*i.e.*, the performance of fine-tuning a pretrained model on a new domain, such as fine-tuning a ImageNet-pretrained model for an object detection task), analogue to Perspective I.
> > >      - In contrast, our method is motivated by Perspective II, *i.e.*, finding a more transferable adversarial example in the loss landscape of a fixed surrogate model.
> > >      - We believe that the above two approaches can be naturally combined to further boost adversarial transferability. The chance of finding a high-transferable adversarial example in a high-transferable surrogate model should be larger than that in a low-transferable surrogate model. It will be explored in our future work.
> > >
> > >
> > > We hope that the above explanations could help to understand our motivation more clearly. Above explanations will be added into our revised manuscript. We are willing to discuss with you any further concerns. Thanks again for this insightful and inspired comment; and it helps us to clarify our motivation more clearly and inspires new directions in our future work.
> > >
> > >
> > >
> > >
> > > [1] Sharpness-aware Minimization for Efficiently Improving Generalization, ICLR 2021.
> > >
> > > [2] Uncovering the Connections Between Adversarial Transferability and Knowledge Transferability, ICML 2021.

---

> > > > ### Comment · Reviewer_PVqn · 2021-11-26
> > > > **Thanks.**
> > > >
> > > > In my original question, what I meant is that to test perspective II "finding a more transferable adversarial example in the loss landscape of a fixed surrogate model", could we consider an experiment to compare with the transferability of adversarial example generated from a normally-trained surrogate model vs. the one with a flatter loss landscape, such as 1) randomized smoothing trained model  https://arxiv.org/abs/1902.02918 or 2) adversarially-trained one https://arxiv.org/abs/2007.08489 (which has better transferability across datasets)?
> > > >
> > > > By doing so, the connection  'model generalization-flatness of loss landscape-attack transferability' can be better motivated.

---

> > > > > ### Author Response · Authors · 2021-11-28
> > > > > **New Reponse**
> > > > >
> > > > > We greatly appreciate your patient clarification and constructive suggestion. The suggested experiments are presented below.
> > > > >
> > > > > **1. Experimental settings**:
> > > > > - **Settings of adversarially trained (AT) model**: Following your suggestion, we adopt the pretrained adversarial training model with $\ell_{2}$ norm [1] as the surrogate model on ImageNet, and the pretrained model is downloaded from the authors' github repository (see https://github.com/Microsoft/robust-models-transfer). We choose the $\ell_{2}$-AT ResNet-50 model with budget 0.05, of which the clean accuracy is 74.78% on ImageNet. For comparison, here we also pick up the results of using the standard ResNet-50 model (with the clean accuracy 76.13% ) as a surrogate model, from Table 4 in the original manuscript.
> > > > > - **Settings of randomized smoothing (RS) trained model**: For randomized smoothing [2], we utilize the pretrained ResNet-50 model with $\sigma$ = 0.25 as the surrogate model. We download it from the authors' github (https://github.com/locuslab/smoothing). Its clean accuracy is 67%, which is the highest among all provided models.
> > > > >
> > > > > - **Others**: We adopt ResNet-50 as the surrogate architecture, while DenseNet-121, VGG-16, and Inception-v3 as target architecture. The *MI-TI-DI* and *MI-TI-DI + RAP-LS* (ours) are adopted as two attack methods.  The hyper-parameters of our RAP-LS are the same as the described in Section 4.1 of our original manuscript, *i.e.*, setting $K_{LS}$ as 100, $\alpha_{n}$ as 2/255, and $\epsilon_{n}$ as 16/255.
> > > > >
> > > > > **2. Results of taking AT model as the surrogate model**: The transfer attack results are shown in Table 1, which can be interpreted from two perspectives:
> > > > >
> > > > > - **Horizontal perspective**: When reading the results in Table 1 horizontally, *i.e.*, given a surrogate model, we compare *MI-TI-DI* and *MI-TI-DI + RAP-LS* on attacking different target models. It is shown that for both surrogate models (standardly trained ResNet-50 in Row 2, adversarially trained ResNet-50 in Row 3), our RAP-LS significantly and consistently improve the attack performance over MI-TI-DI on all target models. It demonstrates the good generalization of our RAP-LS to different types of DNN models.
> > > > >
> > > > > - **Vertical perspective**: When reading the results in Table 1 vertically, *i.e.*, given the same target model, we compare the transfer attack performance between different surrogate models. **1)** It shows that, compared to the standard surrogate model, the AT surrogate model significantly improves the attack performance against Inc-v3 and Dense-121, especially for the MI-TI-DI attack method, while there is a somewhat decrease against VGG-16. It demonstrates that given the same architecture of the surrogate model, the flatness of the loss landscape has a significant effect on the transfer attack performance. It is very likely that the more flatter/smoothed surrogate model has the higher adversarial transferability, but without guarantee, because the architecture difference between surrogate and target models is also an important factor on adversarial transferability. The effects on the adversarial transferability of these factors will be thoroughly studied in our future work. **2) Moreover**, when evaluating the average absolute improvement due to our RAP-LS, it is observed that the improvement is reduced from 18.2% to 9.2%, when replacing the standard by the AT surrogate model. It reveals that our RAP-LS and the smoothed surrogate model have some overlaps on improving the adversarial transferability. The more suitable combination between RAP-LS and the smoothed surrogate model will be further explored in our future work.
> > > > >
> > > > > **3. Results of taking random smoothing (RS) model as surrogate model**:
> > > > > RS model requires adding Gaussian noise multiple times (20-50 times) on each input data and takes the major voting to get its prediction. It leads to much more backward propagations (20-50 times) when generating adversarial examples, which significantly increases the overhead. Attacking the RS model by *MI-TI-DI* and *MI-TI-DI + RAP-LS* takes 35 hours and 75 hours using a Nvidia V100 GPU, respectively. **Our experiments are still in running, and we will report the experimental results immediately after finishing**. Thanks for your understanding and patience.
> > > > >
> > > > > *Table 1: The targeted attack success rate (%). $M^{s}$ stands for surrogate model, and MTDI is the abbreviation of the MI-TI-DI attack method.*
> > > > >
> > > > > |   Attack   |  Inc-v3 | Dense-121 |  VGG-16 | Average Absolute Improvement |
> > > > > | :--------: | :----------: | :----------: | :------: | :------: |
> > > > > |    MTDI/MTDI+RAP-LS ($M^{s}$: ResNet-50)    |    10.9 / 33.2    |      74.9 / 88.5    |    62.8 / 81.5  | 18.2 |
> > > > > |    MTDI/MTDI+RAP-LS ($M^{s}$: ResNet-50-AT-$\ell_{2}$)     |     39.4 / 52.2    |   86.5 / 89.9   |   61.3 / 72.6  | 9.2 |
> > > > >
> > > > > [1] Do Adversarially Robust ImageNet Models Transfer Better? https://arxiv.org/abs/2007.08489
> > > > >
> > > > > [2] Certified Adversarial Robustness via Randomized Smoothing, ICML 2019.

---

> > > > > > ### Comment · Reviewer_PVqn · 2021-11-28
> > > > > > **Thanks and satisfied with the new response**
> > > > > >
> > > > > > Thanks for the follow-up experimental justification. I am satisfied with that and will increase my score. Please be sure to include the above discussion and results in the revised version.

---

> > > > > > > ### Author Response · Authors · 2021-11-29
> > > > > > > **Thanks and all discussions and results will be added**
> > > > > > >
> > > > > > > We are so encouraged by your valued comments and patience. All above discussions and results will be added into the revised manuscript.

---

> ### Author Response · Authors · 2021-11-22
> **Response to Reviewer PVqn (Part 2/2)**
>
> ### **Q4.4**:  "Min-max attack generation could be difficult to tune. Thus, the computation time and hyper-parameter setups of the proposed approach should be clearly stated."
>
>
> ### **R4.4**:
>
> **Hyper-parameter setups**: In the main submission, we have provided some detailed descriptions of the hyper-parameter setting in the third paragraph of Section 4.1 and conducted an ablation study of model parameters in Section 4.7.
>
> **Computation time**: We conducted all experiments in an Nvidia-V100 GPU. Taking MTDI as an example, its computation time is 2 hours. Combining RAP or RAP-LS, the computation time becomes 5.5 or 4 hours. We have added this in our revised version.
>
> ----
>
>
> ### **Q4.5**: "If the n^{adv} is regarded as model noise to flatten the loss landscape, then how does the ...... w.r.t. input rather than model parameters."
>
> ### **R4.5**:
>
> Thanks for this constructive comment.
>
> We followed the suggestion and tried to flatten the loss landscape w.r.t. model parameters by treating $n^{adv}$ as model noise. However, its optimization is much harder than flattening the loss landscape w.r.t. input, because the whole search space of model parameters is much larger than that of input.
>
> Indeed, in our main submission, we have compared two methods that share the similar ideas of flattening the loss landscape w.r.t. the model parameters: LinBP [2] and SGM [3]. They make some modifications to the model parameters to obtain more linear decision boundaries, so as to improve the adversarial transferability.
>
> We have discussed them in Section 2 Related Work and compared our methods with them following their settings in Section 4.5 in the main submission. In LinBP [2], the authors removed the relu in the last two layers and finetuned the whole models for VGG architecture. For ResNet, LinBP and SGM assign the larger weight to skip connection during the gradient back-propagation of generation attacks.
>
> The comparison results are shown in Table 5 in Section 4.5 of the main submission. The results demonstrate that our methods significantly outperform the LinBP and SGM, especially for the targeted attacks. This also demonstrates that the superiority of flatting the loss landscape w.r.t input.
>
> [2] Backpropagating Linearly Improves Transferability of Adversarial Examples, NeurIPS 2020
>
> [3] Skip Connections Matter: On the Transferability of Adversarial Examples Generated with ResNets, ICLR 2020

---

### Official Review · Reviewer_b3AS · 2021-10-31

**Correctness:** 4
**Technical Novelty And Significance:** 4
**Empirical Novelty And Significance:** 4
**Recommendation:** 6
**Confidence:** 4

**Details Of Ethics Concerns:**

I have no ethics concerns.

**Main Review:**

Strengths:
- The writing is clear and the proposed method is easy to follow.
- The idea to seek a flat region of loss landscape to improve the transferability is novel. Instead of purely minimizing the adversarial loss at a single adversarial point, the paper proposes to find a flat region of loss landscape by bi-level max-min optimization, so as to eliminate the overfitting problem of adversarial attacks.
-  The experimental results demonstrated the effectiveness of the proposed method. By injecting the proposed attack method into the existing attack method, the paper improves the transferability of adversarial examples by a large marge in both untargeted attacks and targeted attacks.

Weaknesses:
- The evaluation of defense methods is somewhat limited, the paper only considers the ensemble adversarial training, and many advanced defense methods are lost (such as feature denoising [1] and NRP [2]).
- A drawback of the proposed method is that it requires to conduct T steps update in the inner maximization, which increases the overhead of adversarial attacks.

Suggestions:
- In section 4.2, the paper plots the visualizations of loss landscapes of targeted attacks,  the loss landscapes of untargeted attacks are also needed.

[1] Feature denoising for improving adversarial robustness. CVPR 2019.
[2] A self-supervised approach for adversarial robustness. CVPR 2020.

**Summary Of The Paper:**

This paper focuses on the transferability of the adversarial examples and proposes to boost the transferability by reversing the adversarial perturbation. The motivation is that the flatness of the loss landscape can help to alleviate the overfitting to surrogate models, and thus improve the transferability. Specifically,  instead of purely minimizing the adversarial loss at a single adversarial point, this paper injects the worst-case perturbation for each step of the optimization procedure. Experimental results show that the proposed method surpasses the SOTA transfer-based attack methods by a clear margin, which demonstrates its effectiveness.

**Summary Of The Review:**

Overall, the paper proposes a novel adversarial attack, which is effective and easy to follow. I think the paper is marginally above the acceptance threshold, although there are some weaknesses.

---

> ### Author Response · Authors · 2021-11-22
> **Response to Reviewer b3AS (Part 1/2)**
>
> ### **Q3.1**: "The evaluation of defense methods is somewhat limited, the paper only considers the ensemble adversarial training, and many advanced defense methods are lost (such as feature denoising [1] and NRP [2])."
>
>
> ### **R3.1**:
> Thanks for this valuable comment. We have enriched our experiments by including more advanced defense methods: 1) Feature Denoising, we utilized the pre-trained ResNet-152 model [1] provided by the authors. 2) NRP, we adopted the pre-trained purifiers [2] provided by the authors. 3) Two more multi-step PGD AT models from [3] (the $\ell_{\infty}$ ResNet-50 AT model with budget $4/255$ that ranks first in the RobustBench leaderboard [4] and the $\ell_{2}$ ResNet-50 AT model with budget $0.5$).
> The results of Feature Denoising and the two additional AT models are given in Table 1. For NPR, since it is an offline defense module, we combined it with the two ensemble AT models used in our original submission and the above two additional AT models. The results are given in the following Table 2.
>
> **Experimental Results**: The untargeted attack performance is shown in the below tables. We also follow the experimental settings in Section 4.6 of the main submission. We can observe that our RAP-LS further boosts the attack performance of baseline methods on these defense models. For Feature Denoising and AT models, RAP-LS improves by 5.5\% for the average attack success rate. Combining NRP with AT models is a much stronger defense mechanism, but RAP-LS still achieves the improvement by 0.8%.
>
> *Table 1: The untargted attack success rate (%) on defense models.*
>
>
> |    Attack    | Res-50 AT($\ell_{2}$) | Res-50 AT($\ell_{\infty}$) | Feature Denoising |
> | :----------: | :--------------------------: | :-----------------------------------------: | :------------------: |
> |     MTDI     |             42.5             |                    32.4                     |         44.1         |
> | MTDI+RAP-LS  |             59.5             |                    34.4                     |         44.4         |
> |    MTDSI     |             56.6             |                    35.8                     |         45.0         |
> | MTDSI+RAP-LS |             70.3             |                    36.6                     |         45.7         |
> |    MTDAI     |             62.1             |                    35.6                     |         44.2         |
> | MTDAI+RAP-LS |             73.7             |                    37.7                     |         45.2         |
>
> *Table 2: The untargted attack success rate (%) on defense models against **NRP**.*
>
> |    Attack    | Inc-v3$_{adv}$ | IncRes-v2$_{ens}$ | Res-50 AT($\ell_{2}$) | Res-50 AT($\ell_{\infty}$) |
> | :----------: | :--------: | :----------------: | :--------------------------: | :-----------------------------------------: |
> |     MTDI     |    23.1    |        13.5        |             14.2             |                    25.7                     |
> | MTDI+RAP-LS  |    22.7    |        14.8        |             14.9             |                    26.3                     |
> |    MTDSI     |    22.5    |        14.2        |             15.0             |                    26.1                     |
> | MTDSI+RAP-LS |    24.5    |        15.3        |             15.4             |                    26.2                     |
> |    MTDAI     |    24.1    |        14.7        |             14.2             |                    25.9                     |
> | MTDAI+RAP-LS |    24.9    |        15.6        |             15.3             |                    26.1                     |
>
> [1] https://github.com/facebookresearch/ImageNet-Adversarial-Training
>
> [2] https://github.com/Muzammal-Naseer/NRP
>
> [3] https://github.com/microsoft/robust-models-transfer
>
> [4] https://robustbench.github.io/
>
> ----

---

> ### Author Response · Authors · 2021-11-22
> **Response to Reviewer b3AS (Part 2/2)**
>
> ### **Q3.2**: "A drawback of the proposed method is that it requires to conduct T steps update in the inner maximization, which increases the overhead of adversarial attacks."
>
>
> ### **R3.2**:
>
> Thanks for pointing out this.
> By including the inner maximization process, our method will increase the time cost of adversarial example generation process. However, the adversarial example generation process is conducted based on the offline surrogate models. Compared with this offline time cost, the attacking performance is much more important for black-box attacks, which is also the main advantage of our method. Besides, the late-start strategy could alleviate the time cost and we will explore more in the future.
>
> ----
>
> ### **Q3.3**: About suggestion: "In section 4.2, the paper plots the visualizations of loss landscapes of targeted attacks, the loss landscapes of untargeted attacks are also needed."
>
> ### **R3.3**:
>
> Thanks for this helpful suggestion. We have added visualizations of untargeted attacks in supplementary materials.
>
> ----

---

> ### Comment · Reviewer_b3AS · 2021-11-29
> **Thanks for the detail reponses**
>
> After reading the comments from other reviewers and the responses from the authors, most of my concerns have been addressed. I believe this paper is marginally above the acceptance threshold, and I will keep the original score.

---

### Official Review · Reviewer_5ohk · 2021-11-03

**Correctness:** 3
**Technical Novelty And Significance:** 2
**Empirical Novelty And Significance:** 2
**Recommendation:** 5
**Confidence:** 4

**Main Review:**

The paper is easy to follow and the motivation is clear. However, the technical novelty of the paper is limited. The proposed attack method mainly leverage the minmax framework to search for regions that may have high adversarial transferability loss and then minimize the loss for those regions so as to achieve “flat” loss regions. The algorithms to solve the minmax optimization is standard and follows the existing work. There is no convergence, or any transferability guarantee.

Empirically, from table 3, the targeted attack success rate improvement is usually not significant. Since the paper aims to propose a more transferable attack, it would be interesting to see if it can attack against existing defenses, including the gradient obfuscated ones comparing BPDA attack [1] and adversarially trained models.

[1] Athalye, Anish, Nicholas Carlini, and David Wagner. "Obfuscated gradients give a false sense of security: Circumventing defenses to adversarial examples." International conference on machine learning. PMLR, 2018.


**Summary Of The Paper:**

This paper proposes an adversarial attack, RAP, to boost the transferability of adversarial examples, based on the intuition of flatness of loss landscape and model generalization.
Experimental results show that the proposed attack is more effective against real world APIs.



**Summary Of The Review:**

Overall the paper is well motivated, and the problem statement is clear. However the technical novelty is limited and more baseliens on improving adversarial transferability need to be compared with.
It would be interesting to see if the proposed method can boost the adversarial transferability for different tasks such as object detections as well.

---

> ### Author Response · Authors · 2021-11-22
> **Response to Reviewer 5ohk (Part 1/3)**
>
> ### **Q2.1**: "The paper is easy to follow and the motivation is clear. However, the technical ...... The algorithms to solve the minmax optimization is standard and follows the existing work. There is no convergence or any transferability guarantee."
>
> ### **R2.1**:
>
> **The algorithm to solve the minmax optimization is standard and follows the existing work:**
> We should emphasize that the first contribution of our work is to consider the flatness of loss landscape around the adversarial example, rather than purely minimizing the adversarial loss at a single adversarial point as did in existing works. To achieve this goal, we proposed to minimize the maximal loss value within a local neighborhood region around the adversarial example ${x}^{adv}$, leading to a min-max optimization problem. The solving algorithm of the min-max problem is not our main focus and we thus just adopt the standard algorithm.
>
> **There is no convergence, or any transferability guarantee.**:
> Although we adopt the standard min-max optimization algorithm, we understand that the theoretical convergence or transferability guarantee is always expected. However, to the best of our knowledge, there is still no rigorous convergence proof of non-convex and non-concave min-max problems. Yet, experimentally, we found the algorithm obtains good convergence in our experiments (please see the examples in Figure 1 of the main submission).
> As for the theoretical transferability guarantee, we admit that it is very hard and currently we could only verify the effectiveness of the proposed method experimentally. The consistent improvements and meaningful visualizations of RAP in Figure 2 of the main submission could reveal a strong connection between the flatness of loss landscape w.r.t input and the adversarial transferability. We will further study the theoretical connection between them in the future. Besides, though many transfer attack methods have been proposed in the literature, there is little theoretical support for adversarial transferability. We will do our best to reduce this gap in the future.
> ----
>
> ### **Q2.2**: "Empirically, from table 3, the targeted attack success rate improvement is usually not significant."
>
>
> ### **R2.2**:
>
> Although the targeted attack is more difficult, our method still significantly improves the attack performance on all baseline methods. As shown in Table 3 in the main submission, taking ResNet-50 and DenseNet-121 as surrogate models, the average performance improvements induced by RAP are $5.0$% (I), $8.1$% (MI), $4.6$% (TI), $10.4$% (DI), $18.5$% (SI), and $15.1$% (Admix), respectively. As discussed in Section 4.4 of the main submission, the baseline attacks generally achieve lower transferability when using the VGG-16 or Inception-v3 as surrogate models due to the large difference between ResNet and VGG on model architecture, which has also been verified in existing works [7,8]. However, our methods still achieve relatively large improvements on all baseline methods for these two surrogate models.
>
> ----

---

> > ### Comment · Reviewer_5ohk · 2021-11-28
> > **Post rebuttal comments**
> >
> > Thanks for the detailed answers from the authors. I will keep my score mainly due to the novelty consideration, and I believe the paper would be largely improved by adding the additional comparisons.

---

> > > ### Author Response · Authors · 2021-11-29
> > > **Thanks for your constructive comments, and our thoughts on novelty**
> > >
> > > We are so encouraged by your recognition of our efforts on adding more experiments. They will be added into the revised manuscript.
> > >
> > > We sincerely respect and understand your concern on the (theoretical) novelty. The evaluation of novelty/value/contribution of one work is always the core issue during reviewing. We have to admit that sometimes it is a bit subjective, and it is not surprising that different persons may have different opinions, due to their differences on background or preference. We would like to share our thoughts on this issue with the Reviewers and Area Chair. We think that the value of one work could be evaluated from the following two aspects.
> > >
> > > - **Theoretical aspect**: It is always desired and appreciated that if one work could provide a novel theorem or solid theoretical analysis. Unfortunately, it is often difficult to provide a solid analysis about DNN-based methods, due to the well known properties of DNNs, i.e., black-box, highly nonlinearity. For example, it is still an open question that why the adversarial transferability among DNN models exists. There have been many seminal attempts to reveal the intrinsic reason. To the best of our knowledge, there is still a tough and long journey to achieve the goal of fully understanding the adversarial transferability in theory. However, our method has provided an interesting and empirical evidence that adversarial examples in flat region may be more transferable, which can be understood from the perspective of mitigating the overfitting to surrogate models (If interested, our detailed discussions with the last reviewer PVqn should be helpful to understand this claim). This evidence sheds some light on adversarial transferability, and we expect it could inspire the further solid theoretical analysis and verification.
> > > - **Methodological aspect**: Proposing a suitable method to effectively solve an important and difficult problem is also very valuable. Our method, and many previous methods in this filed (e.g., the compared methods including MI, TI, SI, DI, please refer to Section 4.1 in our manuscript), belong to this type. Although there is no solid theoretical analysis, these methods could provide insights, evidences, and tools to understand or verify adversarial transferability from different perspectives. These valuable explorations could help us to identify more factors related to adversarial transferability, and to gradually approach the intrinsic reason(s).
> > >
> > > **In summary**, in our opinion, most challenging problems are not thoroughly solved in one single work. More commonly, it is solved by different explorations by many researchers with a gradual, continuous and helical manner, including both theoretical and methodological explorations. New empirical methods could inspire more clear theoretical analysis, while new theoretical analysis could guide the development of more advanced methods. They are mutually reinforced, and none of them could be missing. Thus, we believe that our work is a valuable step forward understanding adversarial transferability, and could inspire further theoretical analysis. Moreover, our proposed RAP-LS method is very effective on significantly enhancing the transfer attack performance, and can be naturally combined with many existing techniques. It could be used as a new state-of-the-art baseline by other researchers to develop more advanced attack methods, or to evaluate the robustness of a DNN model using the black-box manner. **We believe that this work is valuable and contributive to the research community of adversarial examples**.

---

> ### Author Response · Authors · 2021-11-22
> **Response to Reviewer 5ohk (Part 2/3)**
>
> ### **Q2.3**: "Since the paper aims to propose ...... including the gradient obfuscated ones comparing BPDA attack [1] and adversarially trained models."
>
> ### **R2.3**:
>
> **The more evaluation on the gradient obfuscated ones and adversarially trained models**: We are a little bit confused about this comment "including the gradient obfuscated ones comparing BPDA attack [1] and adversarially trained models". Since the reviewer talked about the defense methods in the context, we think "including the gradient obfuscated ones comparing BPDA attack" means the defense methods belonging to gradient masking.
> Therefore, we adopted R&P defense [1] and Feature Denoising [2].  We adopted the source code provided by [3] to implement R&P. For the Feature Denoising, we adopt the pre-trained ResNet-152 model [4] provided by the authors.
> For adversarially trained models, we adopt the pre-trained ResNet-50 AT models from [5].  For AT with $\ell_{\infty}$ norm, we adopt the ResNet-50 AT model with the budget $4/255$, which also ranks first in the RobustBench leaderboard [6]. For AT with $\ell_{2}$ norm, we adopt the ResNet-50 AT model with budget $0.5$. For R&P, we combine it with the two ensemble AT models used in our original submission and the above two additional AT models.
>
> **Experimental Results**: The untargeted attack performance is shown in the below tables. We also follow the experimental settings in Section 4.6 of the main submission. We can observe that our RAP-LS significantly boosts the attack performance of baseline methods on these defense models. For R&P defense, RAP-LS achieves the increase by $9.1$ % in terms of average attack success rate. For Feature Denoising and AT models, RAP-LS improves by $5.5$ % for the average attack success rate.
>
> It would be greatly appreciated if the reviewer can provide further comments to help us to understand more correctly.
>
> *Table 1: The untargted attack success rate (%) on defense models.*
>
>
> |    Attack    | Res-50 AT ($\ell_{2}$) | Res-50 AT ($\ell_{\infty}$) |Feature Denoising |
> | :----------: | :--------------------------: | :-----------------------------------------: | :------------------: |
> |     MTDI     |             42.5             |                    32.4                     |         44.1         |
> | MTDI+RAP-LS  |             59.5             |                    34.4                     |         44.4         |
> |    MTDSI     |             56.6             |                    35.8                     |         45.0         |
> | MTDSI+RAP-LS |             70.3             |                    36.6                     |         45.7         |
> |    MTDAI     |             62.1             |                    35.6                     |         44.2         |
> | MTDAI+RAP-LS |             73.7             |                    37.7                     |         45.2         |
>
> *Table 2: The untargted attack success rate (%) against R&P (random resizing and padding).*
>
> |    Attack    | Inc-v3$_{adv}$ | IncRes-v2$_{ens}$ | Res-50 AT ($\ell_{2}$) | Res-50 AT($\ell_{\infty}$) |
> | :----------: | :--------: | :----------------: | :--------------------------: | :-----------------------------------------: |
> |     MTDI     |    65.0    |        46.2        |             52.5             |                    43.7                     |
> | MTDI+RAP-LS  |    82.1    |        63.2        |             65.3             |                    45.8                     |
> |    MTDSI     |    86.5    |        69.6        |             64.1             |                    45.9                     |
> | MTDSI+RAP-LS |    93.4    |        84.9        |             74.0             |                    46.2                     |
> |    MTDAI     |    88.9    |        76.5        |             68.4             |                    46.2                     |
> | MTDAI+RAP-LS |    94.8    |        87.0        |             77.7             |                    47.7                     |
>
> [1] Mitigating Adversarial Effects Through Randomization, ICLR 2018
>
> [2] Feature Denoising for Improving Adversarial Robustness, CVPR 2019
>
> [3] Benchmarking Adversarial Robustness on Image Classification, CVPR 2020
>
> [4] https://github.com/facebookresearch/ImageNet-Adversarial-Training
>
> [5] https://github.com/microsoft/robust-models-transfer
>
> [6] https://robustbench.github.io/
>
> [7] Improving Transferability of Adversarial Examples with Input Diversity, CVPR 2019
>
> [8] On success and simplicity: A second look at transferable targeted attacks, NeurIPS 2021
>
> ----

---

> ### Author Response · Authors · 2021-11-22
> **Response to Reviewer 5ohk (Part 3/3)**
>
>
> ### **Q2.4**: "It would be interesting to see if the proposed method can boost the adversarial transferability for different tasks such as object detections as well."
>
>
> ### **R2.4**:
>
> Thanks for this interesting thought. For attacking object detection models, the adversaries need to get predicted boxes and then optimize the above objective based on current predicted boxes. The predicted boxes vary between consecutive iterations, so the searching space of variables is also different between iterations. Therefore, it is hard to directly apply the existing transfer methods to optimize the objective function of detection. However, we think it is a very interesting topic and will further study it in the future.

---

### Official Review · Reviewer_Mzh9 · 2021-11-05

**Correctness:** 4
**Technical Novelty And Significance:** 3
**Empirical Novelty And Significance:** 3
**Recommendation:** 6
**Confidence:** 4

**Main Review:**

Paper is well written, evaluation is reasonably good and thorough. Results show that method helps to improve transferability of adversarial examples.

Nevertheless there are few things in evaluation that could be improved:

1. Authors use I-FGSM as one of the baseline attacks, but it’s not clear whether they do random restarts (as described in https://arxiv.org/abs/1706.06083 ). It would be useful to clarify this and if authors don’t do random restarts then add an attack with random restarts.

2. Most evaluation is done on undefended models. It would be interesting to study attack performance when source and/or target model is adversarially trained. While authors mentioned few defended models from Tramer at al 2018, it has been shown later that ensemble adversarial training is not particularly strong defence. It’s much better to perform multi-step PGD adversarial training, like in https://arxiv.org/abs/1812.03411. Note that it has been shown that denoising used in https://arxiv.org/abs/1812.03411 is not a good defense, nevertheless authors do produce an adversarially trained model without denoising.

**Summary Of The Paper:**

Paper proposes a novel adversarial transferability attack (i.e. an adversarial attack when a surrogate model is used to attack an unknown model). Proposed method works by modifying iterative proceduce to find adversarial examples, such that it tend to find adversarial examples in flatter regions of a loss surface.
Authors conduct thorough study of the method.


**Summary Of The Review:**

Reasonably good paper. There are few potential improvements for evaluation procedure.

---

> ### Author Response · Authors · 2021-11-22
> **Response to Reviewer Mzh9**
>
> ### **Q1.1**: "Authors use I-FGSM as one of the baseline attacks, but it’s not clear whether they do random restarts ....... with random start"
>
> ### **R1.1**:
> Thanks for this constructive suggestion.
> In the original submission, following the previous works [1,2,3,4], we adopted I-FGSM (without random starts) in our evaluations. For the reviewer's suggestion, we added experiments of I-FGSM with random-start (denoted as I$^*$). The results are shown below.
>
> **Experimental Results.** We followed the experimental settings in Section 4.3 of the main submission. The untargeted attack performance of I$^*$ and I$^*$ combined with RAP is shown in Table 1. RAP is also effective for I-FGSM with random-start. It achieves improvements for I$^*$ on each target model, and with late-start, RAP-LS further enhances the transfer attack performance for I$^*$, achieving $6.0$% improvement in terms of average attack success rate. We have added this comparison to the supplementary materials.
>
> *Table 1: The untargted attack success rate (%). $\mathcal{M}^{s}$ stands for surrogate model.*
>
> |   Attack ($\mathcal{M}^{s}$: Inc-v3)  | \| Res-50 | \| Dense-121 | \| VGG-16 |
> | :--------: | :-------: | :----------: | :------: |
> |    I* / +RAP / +RAP-LS    |   \| 50.9 / 55.2 / 57.9   |     \| 48.6 / 54.0 / 56.2    |   \| 54.7 / 61.6 / 63.8  |
>
>
> |   Attack ($\mathcal{M}^{s}$: Res-50)  | \| Inc-v3 | \| Dense-121 | \| VGG-16 |
> | :--------: | :----------: | :----------: | :------: |
> |    I* / +RAP / +RAP-LS     |    \| 35.0 / 47.2 / 48.9    |     \| 75.7 / 80.2 / 81.5    |   \| 78.0 / 80.5 / 82.1  |
>
>
> |   Attack ($\mathcal{M}^{s}$: Dense-121)  | \| Inc-v3 | \| Res-50 | \| VGG-16 |
> | :--------: | :----------: | :-------: | :------: |
> |    I* / +RAP / +RAP-LS     |    \| 47.0 / 54.2 / 55.1    |  \| 85.6 / 86.1 / 86.9   |   \| 83.7 / 84.6 / 85.0  |
>
>
> |   Attack ($\mathcal{M}^{s}$: VGG-16)  | \| Inc-v3 | \| Res-50 | \| Dense-121 |
> | :--------: | :----------: | :-------: | :----------: |
> |    I* / +RAP / +RAP-LS     |    \| 21.7 / 24.5 / 25.1     |  \| 52.9 / 56.8 / 57.2   |    \| 49.9 / 54.5 / 55.3    |
>
>
>
>
> [1] Improving Transferability of Adversarial Examples with Input Diversity, CVPR 2019
>
> [2] Nesterov accelerated gradient and scale invariance for adversarial attacks, ICLR 2020
>
> [3] Admix: Enhancing the Transferability of Adversarial Attacks, ICCV 2021
>
> [4] On success and simplicity: A second look at transferable targeted attacks, NeurIPS 2021
>
> ----
> ### **Q1.2**: "Most evaluation is done on undefended models ...... nevertheless authors do produce an adversarially trained model without denoising."
>
> ### **R1.2**:
> Thanks for this valuable comment.
> Following the reviewer's suggestion, we added the multi-step PGD adversarial trained models for comparison:
> 1) For the given reference [5], we searched its GitHub project [6] but did not find the adversarially trained model without denoising. Therefore, we utilize its released ResNet-152 Feature Denoise model.
> 2) Apart from [5], for the multi-step PGD AT model on ImageNet, we also adopt the pre-trained ResNet-50 AT models from [7]. For AT with $\ell_{\infty}$ norm, we adopt the ResNet-50 AT model with the budget $4/255$, which also ranks first in the RobustBench leaderboard [8]. For AT with $\ell_{2}$ norm, we adopt the ResNet-50 AT model with budget $0.5$.
>
> **Experimental Results**: The untargeted attack performance is shown in the below table. We also followed the experimental settings in Section 4.6 of the main submission.
> We can observe that our RAP-LS further boosts the transferability of baseline methods on all three defense models, getting the boost by $5.5$% for the average attack success rate. We have added this comparison to the supplementary materials.
>
> *Table 2: The untargted attack success rate (%) on defense models.*
>
>
> |    Attack    | Res-50 AT ($\ell_{2}$)[7] | Res-50 AT($\ell_{\infty}$)[7] | Feature Denoising[5] |
> | :----------: | :--------------------------: | :-----------------------------------------: | :------------------: |
> |     MTDI     |             42.5             |                    32.4                     |         44.1         |
> | MTDI+RAP-LS  |             59.5             |                    34.4                     |         44.4         |
> |    MTDSI     |             56.6             |                    35.8                     |         45.0         |
> | MTDSI+RAP-LS |             70.3             |                    36.6                     |         45.7         |
> |    MTDAI     |             62.1             |                    35.6                     |         44.2         |
> | MTDAI+RAP-LS |             73.7             |                    37.7                     |         45.2         |
>
>
>
> [5] Feature Denoising for Improving Adversarial Robustness, CVPR 2019
>
> [6] https://github.com/facebookresearch/ImageNet-Adversarial-Training
>
> [7] https://github.com/microsoft/robust-models-transfer
>
> [8] https://robustbench.github.io/
>
> ----

---

### Decision · Program_Chairs · 2022-01-20

**Decision:**

Reject

**Comment:**

This paper studies the transferability of adversarial attacks in deep neural networks. In particular, it proposes the reverse adversarial perturbation (RAP) method to boost attack transferability by flattening the landscape of the loss function.

The reviewers acknowledge the strengths of the paper, which include effectiveness of the simple RAP method proposed and the extensive experimentation presented.

However, a number of outstanding concerns still remain. Some of them include the technical novelty of the paper, insufficient theoretical justification of the proposed method, lack of grounded justification between flatness of the loss landscape and model generalization under the specific context of attack transferability, similarity of the optimization problem with some existing work, potential difficulties of the min-max attack generation problem, among others.

As it stands, this is a borderline paper that is reasonably good, but not great. Addressing the outstanding concerns will make the paper more ready for publication in ICLR.